# Domain-Specialized Tree of Thought through Plug-and-Play Predictors

## Abstract

While Large Language Models (LLMs) have advanced complex reasoning, prominent methods like the Tree of Thoughts (ToT) framework face a critical trade-off between exploration depth and computational efficiency. Existing ToT implementations often rely on heavyweight LLM-based self-evaluation or rigid heuristics for branch pruning, making them prohibitively expensive and inflexible for broad application. To address this, we introduce DST, an adaptable, plug-and-play predictor that serves as a lightweight, supervised heuristic to guide the ToT search process. Our predictor enables dynamic, context-aware pruning, allowing the search to proceed with near-greedy efficiency on simpler reasoning steps while adaptively expanding the search beam only when encountering uncertainty or task complexity. We evaluate our approach on a diverse suite of benchmarks spanning mathematical reasoning, general reasoning, and complex logical reasoning. Experimental results demonstrate that our method achieves accuracy competitive with or superior to strong baselines, including standard ToT, while reducing computational overhead by 26-75%. Our work effectively resolves the accuracy-efficiency trade-off in tree-based reasoning, transforming ToT from a resource-intensive technique into a scalable and practical paradigm for complex problem-solving in LLMs.

## 1 Introduction

Large Language Models (LLMs) have demonstrated remarkable reasoning capabilities across diverse domains, ranging from mathematics and programming to planning and scientific discovery. By using chain-of-thought prompting (Wei et al., 2023), tool use (Schick et al., 2023; Gao et al., 2025), and multi-agent collaboration (Wu et al., 2023), recent advances have pushed LLMs beyond simple pattern matching toward complex problem solving. Despite this progress, reasoning with LLMs remains imperfect. Models often produce incorrect intermediate steps, pursue unproductive solution paths, or become trapped in lengthy reasoning chains (Zhang et al., 2023).

Several approaches have been proposed to improve LLM reasoning capability. For post-training methods such as reinforcement learning with human feedback (Schulman et al., 2017; Rafailov et al., 2024; Shao et al., 2024), models are optimized to better follow human preferences. While effective, such approaches are computationally costly, requiring expensive fine-tuning runs. On the other hand, test-time methods enhance reasoning without modifying model parameters. For instance, the Tree of Thoughts (ToT) (Yao et al., 2023) framework extends stepwise reasoning into a tree search, where each partial reasoning step is assigned a score reflecting its promise toward solving the task. The scores are used to determine which nodes to expand and which branches to prune, allowing the model to concentrate its computation on the most promising reasoning paths.

A number of recent works have extended the Tree-of-Thoughts (ToT) paradigm Yao et al. (2023) by incorporating different reasoning guidance. ProbTree Cao et al. (2023) employs probabilistic scoring, while DPTS Ding et al. (2025) leverages confidence estimates and AGoT Pandey et al. (2025) adapts task-specific heuristics. Other variants introduce interactive designs, such as iToT Boyle et al. (2024) with tool-cost awareness and MA-ToT Haji et al. (2024) using validator agents. Preference-based methods have also emerged, including BPP-Search Wang et al. (2025) and CPO Zhang et al. (2024b), as well as retrieval-augmented approaches like RATT Zhang et al. (2024a).

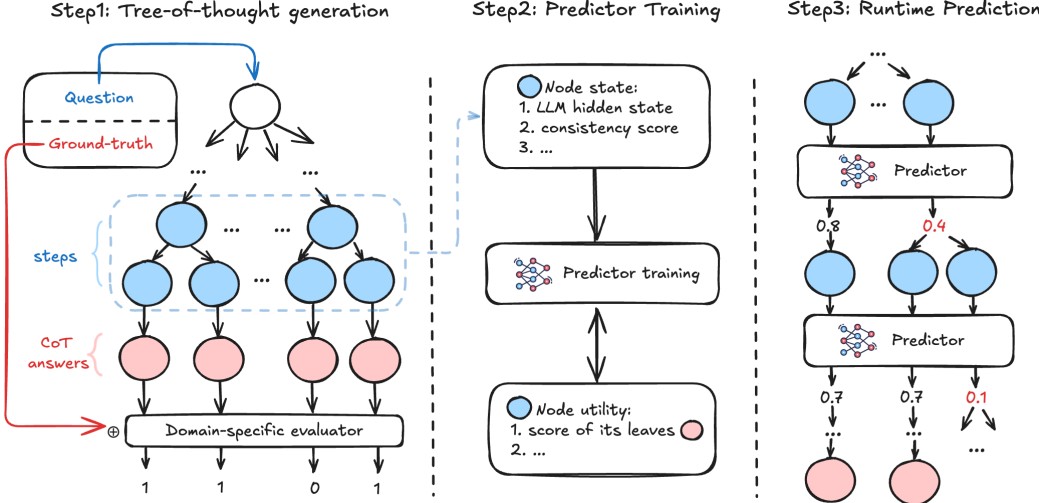

Figure 1: Overview of DST.

However, evaluating whether a partial chain is promising at test time is challenging. First, it must be lightweight, since relying on repeated LLM self-evaluation is prohibitively expensive and introduces significant computational overhead (Madaan et al., 2023). Second, it should be easily adaptable to new domains; methods that depend on manually crafted rules or rigid, task-specific verifiers lack such flexibility and require extensive engineering efforts (Gao et al., 2023). Finally, it must effectively forecast the potential utility of partial reasoning chains. Prior work such as DPTS (Ding et al., 2025) relies on local confidence scores to guide parallel expansion. However, confidence alone does not necessarily predict the future utility of a reasoning path, since confident steps may still lead to hallucinations or unproductive exploration.

To address the above challenges, we propose DST, which extends ToT framework by introducing a novel adaptable plug-and-play predictor that enables efficient control over the ToT search process. As illustrated in Figure 1, the predictor serves as a heuristic, supervised scorer, making immediate, context-aware judgments for branch selection at each step in the search. Specifically, at each search step, our predictor evaluates the initial generated thought and assigns it a confidence score. If this score exceeds a predefined threshold, the system commits to this "good-enough" path greedily, effectively behaving like a single-chain reasoner and avoiding the cost of generating further alternatives. Conversely, if the score falls below the threshold, indicating uncertainty or a complex decision point, the system dynamically expands the search to a full beam, preserving the robust exploration and error-correction capabilities of traditional ToT.

We validate our approach on several reasoning challenges—including mathematical reasoning (MATH500 (Lightman et al., 2023), GSM8K (Cobbe et al., 2021), Minerva-Math (Lewkowycz et al., 2022), SVAMP (Patel et al., 2021)), general reasoning (GPQA (Rein et al., 2023)), and complex logical reasoning (BBEH (Kazemi et al., 2025)) using state-of-the-art LLMs. Results confirm that our method achieves accuracy competitive with or superior to standard ToT baselines while reducing token consumption by 26-75%. In summary, our work transforms ToT reasoning from an efficiency bottleneck into a fast, widely deployable paradigm, making structured search feasible anywhere LLM inference is used.

Key highlights of our approach:

- Efficiency. The predictor prunes unpromising branches during search, reducing token costs by 26-75% while maintaining or even increasing accuracy on popular benchmarks.

- Plug-and-Play & Domain-General. The predictor is decoupled from the backbone LLM, requiring only lightweight domain-specific training on a small dataset, making it easily transferable across various domains such as math, general QA, and program synthesis.

- Adaptive search. DST dynamically adjusts its search breadth based on the predictor's real-time confidence.

- Test-Time Scalability. Our method drastically reduces the computational overhead of tree-based reasoning at test-time. By replacing expensive LLM-based evaluators with a lightweight predictor, we lower the token consumption per inference run by 26-75%.

## 2 BACKGROUND

LLMs have progressed significantly in their problem-solving capabilities through evolving prompting techniques. *Input-output (IO) prompting* serves as the simplest approach, directly mapping inputs to outputs using few-shot or zero-shot examples. To enhance reasoning performance, *CoT prompting* (Wei et al., 2023) introduces intermediate reasoning steps, enabling the model to decompose complex problems. Building on this, *Self-Consistency (CoT-SC)* (Wang et al., 2023) samples multiple CoT reasoning paths and selects the most frequent answer, improving reliability through ensemble effects. Going beyond linear reasoning, *ToT* framework generalizes CoT by modeling problem solving as a tree search over discrete thoughts, enabling deliberate planning, exploration of alternative solutions, and backtracking. This structured reasoning approach significantly boosts performance on tasks requiring strategy, foresight, and creativity.

Formally, the ToT framework models problem solving as a search over a tree $\mathcal{T}$. Given an input problem $x$, ToT framework initializes a root node $s_0 = (x, \emptyset)$ with an empty thought sequence. During execution, the language model $p_\theta$ dynamically expands the tree through iterative branching: at each node $s = (x, \mathcal{Z})$, the *thought generator* $G(p_\theta, s, k)$ operates on the current state $s' = (x, \mathcal{Z})$ to produce $k$ candidate next thoughts $\{z^{(1)}, \ldots, z^{(k)}\}$, where each thought $z^{(i)}$ extends the existing sequence $\mathcal{Z}$ to form a new state $s^{(i)} = (x, [\mathcal{Z}; z^{(i)}]$. The *state evaluator* $V(p_\theta, s^{(i)})$ then scores these new states, after which a search algorithm selects the most promising node for expansion based on heuristic scores. This process continues until termination criteria are met, ultimately yielding an optimal solution path $\mathcal{Z}^* = \langle z_1^*, \ldots, z_T^* \rangle$ as a chain of thoughts from root to leaf.

The critical bottleneck in this workflow lies with the state evaluator. In the original ToT work, the evaluator relies on expensive LLM self-reflection, which involves prompting the model to critique its own outputs. This introduces substantial computational overhead, making the process impractical for many applications. This motivates us to replace this costly evaluator with a lightweight, pretrained predictor that enables an adaptive search strategy.

**Example.** We illustrate our method with the following problem: "Janet has 5 apples. She buys 2 more boxes of apples, with 6 apples in each box. How many apples does she have in total?"

First, the thought generator produces candidate steps. Candidate (1): "First, calculate the total apples in the boxes. 2 boxes * 6 apples/box = 12 apples." Instead of asking the LLM to reflect, our predictor instantly analyzes key characteristics of this thought and assigns it a score of 0.91. Since 0.91 exceeds our predefined threshold ($\tau = 0.7$), the system triggers a shortcut. It immediately accepts this step and proceeds to the next depth, skipping the generation and evaluation of any alternative candidates for this step. The process at this node becomes as efficient as a single greedy generation. Now, consider a more ambiguous step where the predictor is less certain. The system generates the first candidate: "Calculate 2 times 6..." with predicated score 0.65, which is below the threshold $\tau = 0.7$, so the system cannot take the shortcut, it must continue exploring. Then it generates the next candidate "The total is 5 + 2 * 6..." $\rightarrow$ with predicted score 0.62. It continues this process. If none of the candidates met the shortcut criterion, DST reverts to a full-beam search mode. It collects all generated candidates and expand them in parallel in the next step.

This adaptive mechanism stands in stark contrast to baselines relying on LLM self-reflection, which require generating verbose critiques for every candidate. Our predictor enables immediate, data-driven decisions, maximizing efficiency by exiting early when confident, while retaining the robustness of a full tree search when uncertain. As demonstrated in section 4, this dynamic strategy reduces computational overhead by 26-75% while maintaining or even improving solution accuracy.

## 3 METHOD

DST enhances LLM reasoning through guided search over a ToT framework. Our key innovation is a lightweight **runtime predictor** that serves as an adaptive state evaluator $V(p_\theta, s^{(i)})$. The system operates in two phases: (1) an efficient offline training phase where the predictor is trained on a relatively small set of generated reasoning paths to assess thought quality, and (2) an online inference phase where the predictor dynamically guides the LLM by pruning low-potential branches and expanding promising thought sequences in real-time. This approach enables more efficient and targeted problem-solving compared to conventional reasoning methods.

### 3.1 STATE DEFINITION

We formally define the state by extending a reasoning node in the ToT as a 3-tuple $s = (x_s, \mathcal{Z}_s, \phi_s)$, where $\phi_s = (\mathbf{v}_s, c_s)$ is the feature vector encoding the node's properties. The semantic representation $\mathbf{v} \in \mathbb{R}^d$ is derived from the language model's hidden states via

$$\mathbf{v}_s = \mathbf{h}(p_\theta([x_s; \mathcal{Z}_s]))$$

where $p_\theta([x; \mathcal{Z}])$ denotes the forward pass of the LLM on the concatenated input and reasoning path, and $\mathbf{h}(\cdot)$ extracts the hidden state (e.g., via pooling or [CLS] token embedding). The consistency score $c$ measures the node's alignment with its reasoning history by computing the average similarity between $\mathbf{v}(s)$ and the embeddings of its ancestor states $\mathcal{A}_s = \{s_1, \ldots, s_k\}$ along the path from the root:

$$c_s = \frac{1}{|\mathcal{A}_s|} \sum_{s_i \in \mathcal{A}_s} \text{sim}(\mathbf{v}_s, \mathbf{v}_{s_i})$$

where $\text{sim}(\cdot, \cdot)$ is defined using cosine similarity. This feature operationalizes cognitive coherence, helping the predictor identify and penalize logically disjointed reasoning paths. Together, these features provide the predictor with a comprehensive real-time signal about the semantic content and logical integrity of a reasoning step. Noted that computational cost is not treated as an input feature. Instead, we incentivize efficiency directly within the predictor's training objective. As detailed in subsection 3.2, the ground-truth scores assigned to nodes are recursively discounted by a factor $\gamma$. This implicitly teaches the predictor to favor shorter, more direct paths to a correct solution, as deeper nodes are inherently assigned lower maximum scores. This design embeds a preference for efficiency into the learned value function itself, rather than relying on it as an explicit input feature.

This feature design allows the predictor to evaluate the intrinsic quality of a reasoning state. The semantic vectors $\mathbf{v}_s$ target semantic fidelity, capturing nuanced contextual meaning, while the consistency score $c_s$ enforces logical integrity by penalizing breaks in the reasoning flow.

### 3.2 TRAINING DOMAIN-SPECIALIZED PREDICTOR

**Data collection.** A key advantage of our approach is the lightweight nature of the predictor's training. A central challenge in enhancing reasoning is the difficulty of defining a reward signal for each intermediate thought. Our primary contribution in this area is a process that automatically labels the reward for each node in the thought tree, transforming raw reasoning paths into quantifiable supervision signals. This process, formalized in 1, is designed to efficiently generate a high-quality training set from a relatively small number of initial problems (the specific data splits are detailed in Appendix B). The generation of this training data follows a structured three-phase approach: (1) **breadth-first tree construction** to explore the solution space, (2) **leaf node verification** to establish ground-truth outcomes, and (3) **recursive score propagation** to assign credit to intermediate steps.

First, the **breadth-first tree construction** phase initiates with the input question as the root node, progressively expanding the reasoning space through systematic exploration. At each non-terminal node, the language model generates $k$ potential next steps. Each step is simply formed by generating text until a specific stop criterion is encountered (such as text "#step"). During generation, the algorithm captures the contextual hidden states from the transformer, which form the basis for the feature vector $\phi_s$. As defined previously, this vector includes a semantic representation $\mathbf{v}_s$ derived from these hidden states and a consistency score $c_s$ measuring alignment with the reasoning path. This provides a rich, quantitative signal for the predictor to learn from. The queue-based implementation maintains balanced depth exploration, preventing the path bias inherent in depth-first approaches while ensuring comprehensive coverage of potential solution trajectories.

---

**Algorithm 1** Training Data Collection for Predictor

---

**Require:** LLM $p_\theta$, Input $x$, max depth $d_{max}$, branching factor $k$, discount factor $\gamma$
**Ensure:** Training set $\mathcal{D}$
 1: Initialize root node with state $s_0 \leftarrow (x, \emptyset, null)$
 2: Initialize empty tree $\mathcal{T} \leftarrow \{s_0\}$
 3: Initialize queue $Q \leftarrow [s_0]$
 4: Initialize training set $\mathcal{D} \leftarrow \emptyset$
 5: **while** $Q$ is not empty **do** $\qquad\qquad\qquad\qquad\qquad\qquad\qquad\qquad$ ▷ Tree Construction
 6: $\quad$ $s \leftarrow Q.\text{dequeue}()$
 7: $\quad$ **if** $\text{depth}(s) \geq d_{max}$ **then**
 8: $\quad\quad$ Continue
 9: $\quad$ **end if**
10: $\quad$ Generate $k$ thoughts $\{z^{(1)}, \ldots, z^{(k)}\} \sim p_\theta(\cdot|s)$
11: $\quad$ **for** each candidate $z^{(i)}$ **do**
12: $\quad\quad$ Construct new node with state $s' \leftarrow (x, \mathcal{Z}_s \cup \{z^{(i)}\}, \phi_{s'})$
13: $\quad\quad$ Compute representation $\phi_{s'} \leftarrow [\mathbf{v}_s; c_s]$
14: $\quad\quad$ $\mathcal{T}.\text{add}(s')$
15: $\quad\quad$ $Q.\text{enqueue}(s')$
16: $\quad$ **end for**
17: **end while**
18: $\mathcal{L} \leftarrow \{s \in \mathcal{T} \mid \text{is\_leaf}(s)\}$
19: **for** each $s_l \in \mathcal{L}$ **do** $\qquad\qquad\qquad\qquad\qquad\qquad\qquad$ ▷ Chain-of-Thought Evaluation
20: $\quad$ $y_l \leftarrow \mathbb{I}(\text{is\_correct}(s_l))$
21: **end for**
22: **for** each $s_n \in \text{postorder}(\mathcal{T})$ **do** $\qquad\qquad\qquad\qquad\qquad$ ▷ Score Propagation
23: $\quad$ $y_n \leftarrow \gamma \cdot \frac{1}{|S_c|} \sum_{s \in S_c} y_s$ $\qquad\qquad$ ▷ Apply discounted average of children scores
24: $\quad$ $\mathcal{D} \leftarrow \mathcal{D} \cup \{(\phi_{s_n}, y_n)\}$
25: **end for**
26: **return** $\mathcal{D}$

---

Second, the **leaf nodes verification** phase subjects all terminal nodes to rigorous, domain-appropriate validation. For closed-domain problems with unambiguous solutions, we employ pattern matching against canonical answer formats. Subjective or open-ended tasks utilize natural language inference models to assess answer validity based on semantic entailment. Mathematical reasoning branches leverage symbolic execution engines for programmatic verification. Each terminal node $s_l$ receives a definitive quality assessment $y_l \in 0, 1$, establishing unambiguous ground truth labels that anchor the subsequent scoring framework. This binary labeling provides the foundational signal for the recursive score propagation process.

Finally, the **score propagation** phase assigns a quality score to each non-terminal node in a bottom-up manner. This process begins by calculating the depth of each node, after which nodes are processed in descending order of depth. For any internal node $s_i$, its score $y_i$ is formulated as the average of its children's scores, scaled by a discount factor $\gamma$ (e.g., 0.99). This is formalized by the equation:

$$y_i = \gamma \cdot \frac{1}{|S_c|} \sum_{s \in S_c} y_s \tag{1}$$

where $S_c$ denotes the set of all direct children of node $s_i$ and $y_s$ denotes their scores. This formulation serves two primary functions. First, by averaging the scores of its children, it synthesizes the expected quality of all paths originating from the node, preventing the overestimation of a node's potential due to a few outlier high-quality paths. Second, the discount factor $\gamma$ imposes a penalty on longer reasoning chains, thereby incentivizing the discovery of more concise and efficient solutions. Through this recursive score assignment, each internal node's value comes to accurately reflect its aggregate potential for guiding the model toward a valid conclusion, providing a robust and information-rich supervision signal for training the predictor.

The resulting training set $\mathcal{D} = \{\phi, \mathbf{y}\}$ comprises feature-label pairs spanning all tree nodes, capturing the complete spectrum of reasoning quality from fundamental errors to optimal solution paths. This supervision signal enables the predictor to learn nuanced quality estimation that assesses the

---

**Algorithm 2** ToT with DST for pruning

---

**Require:** Trained predictor $Predict$, input $x$, beam width $b$, max depth $d_{max}$, threshold $\tau$
**Ensure:** Chain of Thought $\pi^*$ or $\emptyset$
1: Initialize beam $\mathcal{B} \leftarrow [\text{root}(x)]$
2: Initialize CoT $\pi^* \leftarrow \emptyset$
3: **for** $t = 1$ **to** $d_{max}$ **do**
4:      Initialize next beam $\mathcal{B}' \leftarrow \emptyset$
5:      **for** each node $s \in \mathcal{B}$ **do**
6:          Generate the first thought $z^{(1)} \sim p_\theta(\cdot|s)$
7:          $s' \leftarrow \text{create\_node}(s, z^{(1)})$
8:          $\phi_{s'} \leftarrow [\mathbf{v}_{s'}; c_{s'}]$
9:          $p_{s'} \leftarrow Predict(\phi_{s'})$                 $\triangleright$ Predict correctness of the first thought
10:          **if** $p_{s'} \geq \tau$ **then**                  $\triangleright$ Early-exit: first thought is good enough
11:             $\mathcal{B}' \leftarrow \mathcal{B}' \cup \{s'\}$
12:             **continue**             $\triangleright$ Prune all other siblings and move to the next node in $\mathcal{B}$
13:          **end if**
14:          Generate $k - 1$ more thoughts $\{z^{(2)}, \ldots, z^{(k)}\} \sim p_\theta(\cdot|s)$    $\triangleright$ Fallback: first thought was not good
15:          $\mathbb{Z} \leftarrow \{z^{(1)}, \ldots, z^{(k)}\}$
16:          **for** each thought $z^{(i)} \in \mathbb{Z}$ **do**
17:             $s_{\text{new}} \leftarrow \text{create\_node}(s, z^{(i)})$
18:             $\phi_{s_{\text{new}}} \leftarrow [\mathbf{v}_{s_{\text{new}}}; c_{s_{\text{new}}}]$
19:             $p_{s_{\text{new}}} \leftarrow Predict(\phi_{s_{\text{new}}})$
20:             **if** $p_{s_{\text{new}}} \geq \tau$ **then**
21:                $\mathcal{B}' \leftarrow \mathcal{B}' \cup \{s_{\text{new}}\}$
22:             **end if**
23:          **end for**
24:      **end for**
25:      $\mathcal{B} \leftarrow \text{top\_b}(\mathcal{B}')$                         $\triangleright$ Select top $b$ nodes by score $p$
26:      **if** $\mathcal{B} = \emptyset$ **then**
27:          **return** $\emptyset$                       $\triangleright$ No valid paths remain
28:      **end if**
29: **end for**
30: Let $s^\star$ be the node in $\mathcal{B}$ with the highest score $p_s$
31: $\pi^\star \leftarrow \text{path\_of}(s^\star)$              $\triangleright$ Select the best leaf node using stored scores
32: **return** $\pi^\star$

---

relative utility of partial solutions while naturally handling class imbalance through score propagation.

### 3.3 PREDICTOR AS RUNTIME EVALUATOR

During the inference phase, we leverage the trained predictor to dynamically control the search strategy, as formalized in Algorithm 2. The process is centered on a predict-first-thought mechanism that balances greedy efficiency with robust beam-search exploration. During each node expansion, the system first generates a single candidate thought $z^{(1)}$. The predictor immediately evaluates this thought, yielding a quality score $p$. This score is then compared against a predefined confidence threshold $\tau$, which acts as a dynamic switch for the search strategy. If the score is high, the system accepts this thought, prunes all potential siblings, and proceeds with single-chain efficiency. If the score is low, indicating uncertainty, the system generates the remaining $b - 1$ candidate thoughts to complete a full beam of size $b$. All candidates, including those with scores below $\tau$, are added to a pool for ranking.

After expanding all nodes at the current depth, the system selects the top-$b$ candidates from the collective pool to form the next beam. Finally, upon reaching the maximum depth $d_{max}$, the path $\pi^\star$ terminating in the leaf node with the highest predictor score is chosen as the final output, ensuring methodological consistency between search and selection.

**Complexity analysis of pruning.** The efficiency gain can be formally analyzed by considering the search space complexity. In a standard ToT framework with a branching factor of $k$ and a maximum depth of $d$, the total number of nodes in the search tree grows exponentially, with a complexity of

$O(k^d)$. Our adaptive method operates at each depth level d with an effective beam width $b_{eff}$. The value of $b_{eff}$ is determined by the predictor's confidence relative to the threshold $\tau$. If the score of the first generated thought $s'$ is higher than $\tau$, the system prunes all other potential siblings. The effective beam width at this step becomes $b_{eff} = 1$. This occurs with probability $\mathbb{P}(p \geq \tau)$. If the score is low, the system falls back to generating the full beam of $b$ candidates to ensure no promising path is missed. The effective beam width at this step is $b_{eff} = b$. This occurs with probability $1 - \mathbb{P}(p \geq \tau)$. The expected effective beam width, $E[b_{eff}]$, at any given step can be modeled as:

$$E[b_{eff}] = 1 \cdot \mathbb{P}(p \geq \tau) + b \cdot (1 - \mathbb{P}(p \geq \tau)) \tag{2}$$

The overall search complexity is then determined by this expected effective branching factor at each depth level. Assuming $\mathbb{P}(p \geq \tau)$ is roughly constant, the complexity of our pruned search tree becomes $O(E[b_{eff}]^d)$, which is significantly lower than the standard beam search complexity of $O(k^d)$. This analysis shows that the efficiency gain is directly controlled by $\mathbb{P}(p \geq \tau)$. When the predictor is confident (high $\mathbb{P}(p \geq \tau)$), $E[b_{eff}]$ approaches 1, and the search approximates CoT. When the predictor is uncertain (low $\mathbb{P}(p \geq \tau)$), $E[b_{eff}]$ approaches b, and the inference retains the robustness of full ToT search.

## 4 EXPERIMENT

### 4.1 MAIN RESULT

**Experimental Setup.** We evaluate our approach using Qwen3-8B (Yang et al., 2025), Llama3.1-8B-Instruct (Grattafiori et al., 2024) and Gemma3-12B-it (Team et al., 2025) as the backbone model across diverse benchmarks spanning mathematical reasoning (GSM8K (Cobbe et al., 2021), SVAMP (Patel et al., 2021), Minerva-math (Lewkowycz et al., 2022), MATH-500 (Lightman et al., 2023)), general reasoning (GPQA (Rein et al., 2023)), and complex logical reasoning tasks (BIG-Bench Extra Hard (Kazemi et al., 2025) subtasks: BoardgameQA (Kazemi et al., 2023), Boolean Expressions, Causal Understanding (Nie et al., 2023; Kıcıman et al., 2024), and Geometric Shapes (Suzgun et al., 2022)). These benchmarks were specifically selected because they are reasonably complex, often requiring multi-step planning and exploration that challenge simpler single-path reasoning methods, making them ideal for assessing the efficacy of non-trivial ToT frameworks. We

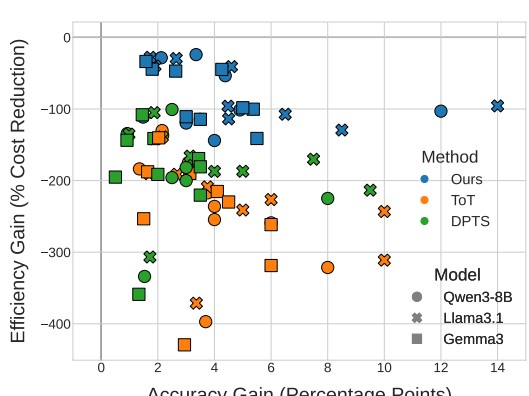

Figure 2: Accuracy vs. Efficiency Trade-off. Each point represents the performance of a method on a specific task and model, plotted as accuracy gain (percentage points) versus efficiency gain (percentage cost reduction) relative to CoT.

compare against three key baseline approaches: (1) Chain-of-Thought prompting (Original CoT), (2) standard Tree-of-Thoughts with LLM-based evaluation (ToT), and (3) Dynamic Parallel Tree Search (DPTS), a recent adaptive ToT variant.

Performance is measured across two primary dimensions: solution accuracy (percentage of correctly solved problems) and computational efficiency (average token consumption per problem). All experiments use identical hardware configurations and temperature settings to ensure fair comparison across methods. Detailed experiment settings can be found in Appendix B.

**Efficiency-Accuracy Trade-off Achievement.** The trade-off between accuracy and efficiency is visualized in Figure 2, with detailed results provided in Table 1. The figure plots the accuracy gain over CoT against the corresponding change in computational cost. Our method consistently populates the upper region of the plot, demonstrating a superior efficiency-accuracy frontier compared to standard ToT and DPTS. This illustrates our method's ability to achieve substantial accuracy improvements without the excessive computational overhead typical of other tree-search methods. A detailed breakdown across task categories reveals the robustness of this behavior.

Table 1: Comparison of Tree-of-Thought reasoning methods. Best performance for each metric (highest accuracy, lowest cost) is shown in **bold**. Results are relative improvements over CoT.

(a) Part I: Mathematical Reasoning (GSM8K, SVAMP, Minerva, MATH-500).

| Model | Method | GSM8K | | SVAMP | | Minerva | | MATH-500 | |
|---|---|---|---|---|---|---|---|---|---|
| | | Acc↑ | Cost↓ | Acc↑ | Cost↓ | Acc↑ | Cost↓ | Acc↑ | Cost↓ |
| Qwen3-8B | CoT | 89.09 | 799.7 | 75.76 | 2125 | 26.80 | 3320 | 92.05 | 2680 |
| | ToT | **+3.69** | +3175 | +1.35 | +3903 | +3.08 | +5808 | +2.15 | +3488 |
| | DPTS | +1.53 | +2670 | +0.92 | +2855 | +2.17 | +4540 | **+2.50** | +2699 |
| | DST | +3.35 | **+192.3** | **+1.48** | +2365 | **+4.38** | **+1776** | +2.12 | **+762** |
| Llama3.1 | CoT | 87.52 | 817 | 76.21 | 2257 | 25.95 | 3416 | 93.55 | 2783 |
| | ToT | **+3.36** | +3033 | +1.69 | +4294 | +2.68 | +6530 | **+2.16** | +3869 |
| | DPTS | +1.72 | +2506 | +0.98 | +3049 | +1.88 | +4839 | +1.87 | +2924 |
| | DST | +2.65 | **+241** | **+1.90** | **+902** | **+4.60** | **+1394** | +1.73 | **+774** |
| Gemma3 | CoT | 93.21 | 850 | 79.53 | 2504 | 31.40 | 3801 | 95.52 | 3107 |
| | ToT | **+2.94** | +3650 | +1.64 | +4702 | +3.13 | +7222 | **+2.03** | +4358 |
| | DPTS | +1.33 | +3050 | +0.91 | +3597 | +1.86 | +5370 | +1.45 | +3359 |
| | DST | +2.63 | **+400** | **+1.80** | **+1111** | **+4.26** | **+1700** | +1.58 | **+1045** |

(b) Part II: General and Logical Reasoning (GPQA, BBEH).

| Model | Method | GPQA | | BoardgameQA | | Boolean | | Causal | | Geo | |
|---|---|---|---|---|---|---|---|---|---|---|---|
| | | Acc↑ | Cost↓ | Acc↑ | Cost↓ | Acc↑ | Cost↓ | Acc↑ | Cost↓ | Acc↑ | Cost↓ |
| Qwen3 | CoT | 44.80 | 4089 | 34.00 | 4425 | 24.00 | 4977 | 42.50 | 4406 | 45.00 | 3468 |
| | ToT | +3.76 | +8875 | +8.00 | +14220 | **+4.00** | +11751 | **+4.00** | +11212 | **+6.00** | +8984 |
| | DPTS | +3.27 | +7001 | +8.00 | +9953 | +3.00 | +9955 | +2.50 | +8627 | +3.00 | +6299 |
| | DST | **+4.90** | **+4141** | **+12.00** | **+4560** | +3.00 | **+5953** | +3.50 | **+5044** | +4.00 | **+4994** |
| Llama3.1 | CoT | 44.06 | 4156 | 31.50 | 4501 | 18.00 | 5055 | 37.00 | 4458 | 25.50 | 3556 |
| | ToT | +3.75 | +8678 | +10.00 | +15004 | **+6.00** | +12450 | +5.00 | +10749 | **+10.00** | +8648 |
| | DPTS | +3.14 | +6892 | +9.50 | +9604 | +5.00 | +9452 | +4.00 | +8343 | +7.50 | +6050 |
| | DST | **+4.48** | **+3994** | **+14.00** | **+4307** | +4.50 | **+5752** | **+6.50** | **+4786** | +8.50 | **+4601** |
| Gemma3 | CoT | 48.13 | 4926 | 33.00 | 5311 | 25.50 | 6010 | 49.00 | 5257 | 32.50 | 4115 |
| | ToT | +4.10 | +10602 | **+6.00** | +16924 | **+4.50** | +13817 | +1.50 | +13319 | **+6.00** | +10762 |
| | DPTS | +3.44 | +8336 | +3.50 | +11701 | +2.00 | +11501 | +0.50 | +10258 | +3.50 | +7437 |
| | DST | **+5.37** | **+4940** | +5.00 | **+5218** | +3.50 | **+6874** | **+3.00** | **+5825** | +5.50 | **+5809** |

On mathematical reasoning tasks, DST provides a highly cost-effective path to performance gains. For instance, on the challenging GSM8K benchmark, DST consistently matches or closely approaches the accuracy of ToT while requiring only about a quarter of the additional token overhead. This efficiency is critical for deploying advanced mathematical reasoning at scale. In the domain of general and logical reasoning, DST's advantages become even more pronounced. On complex benchmarks like GPQA and BoardgameQA, our method frequently outperforms ToT not only in efficiency but also in absolute accuracy. For example, using the Llama3.1 model on BoardgameQA, DST achieves a remarkable +14.00% accuracy improvement over CoT, significantly surpassing ToT's +10.00% gain, yet it does so while consuming less than one-third of the tokens. This highlights DST's capability to navigate complex search spaces more effectively than its expensive counterparts.

A key observation is the consistency of our method's benefits across all three backbone models: Qwen3, Llama3.1, and Gemma3. The core advantage, substantial efficiency gains for a minimal or even positive impact on accuracy, is universal. Whether on Qwen3, Llama3.1, or the more capable Gemma3, our approach consistently delivers token savings in the 26-75% range compared to standard ToT, validating the robustness of our predictor-guided pruning strategy. The universal and

Table 2: Impact of State Feature Components on GSM8K and GPQA. Best performance for each metric (highest accuracy, lowest token cost) is shown in **bold**.

| Method | GSM8K | | GPQA | |
|---|---|---|---|---|
| | Accuracy (%) | Avg. Tokens | Accuracy (%) | Avg. Tokens |
| DST | **92.4** | **992** | **49.7** | **8230** |
| DST w/o $c_s$ (semantics only) | 90.1 | 1150 | 47.0 | 8500 |
| DST w/o $\mathbf{v}_s$ (consistency only) | 85.7 | 1300 | 42.3 | 9800 |

dramatic reduction in computational cost makes our method a more practical and scalable choice across all tested models and tasks.

## 4.2 ABLATION STUDY

### 4.2.1 IMPACT OF STATE FEATURE COMPONENTS

**Experimental Setup.** The experiment begins with our full model, DST-Full, as the baseline. We then systematically disable each feature component to create two variants:

- w/o $c_s$: The model operates without the consistency score, relying only on semantic representation ($\mathbf{v}_s$).

- w/o $\mathbf{v}_s$: The model operates without the semantic representation ($\mathbf{v}_s$), using only consistency ($c_s$).

The results in Table 2 confirm that both feature components are vital. Removing the consistency score (w/o $c_s$) leads to a 2%-3% point accuracy drop and increased token usage, suggesting the model explores less coherent paths. Removing the semantic vector (w/o $\mathbf{v}_s$) is even more detrimental, causing a significant 5%-7% point accuracy loss, as the predictor loses its core understanding of the reasoning content. The full model synergistically combines both signals for the best performance.

### 4.2.2 SENSITIVITY TO HYPERPARAMETERS

We analyzed the model's sensitivity to three key hyperparameters: beam width $b$, pruning threshold $\tau$, and discount factor $\gamma$. Full details of this analysis, including figures, are provided in Appendix C.

Our experiments reveal that a modest beam width ($b = 3$) substantially improves accuracy over a greedy search ($b = 1$), but further increases yield diminishing returns at a high computational cost, which motivates our adaptive search strategy. The pruning threshold $\tau$ is shown to effectively control the accuracy-efficiency trade-off, with performance gains saturating at higher $\tau$ values. Finally, we determined that a slight penalty against verbosity is optimal, with a discount factor of $\gamma = 0.99$ outperforming both unconstrained generation $\gamma = 1.00$ and overly aggressive penalties. These findings validate our default hyperparameter settings.

## 5 CONCLUSION

In this work, we introduced the Domain-Specialized Tree of Thought (DST) framework to resolve the critical efficiency bottleneck in tree-based reasoning. Our core innovation is a lightweight, plug-and-play predictor that is domain-specialized through focused training on a small set of task-specific examples. This predictor replaces the prohibitively expensive, recursive LLM-based evaluators used in standard ToT, enabling an adaptive search that prunes unpromising paths with minimal computational cost. This approach directly translates into significant resource savings, yielding a 26-75% reduction in token consumption while maintaining or even improving accuracy over baseline methods. By decoupling the search heuristic from the main LLM, DST transforms structured reasoning from a resource-intensive technique into a scalable and practical paradigm, making sophisticated problem-solving economically viable for real-world applications where computational efficiency is a primary constraint.

## ETHICS STATEMENT

All authors of this paper adhere to the ICLR Code of Ethics. Our work is primarily algorithmic, focusing on enhancing the computational efficiency of reasoning systems in Large Language Models (LLMs). The research relies on publicly available datasets and open-source pre-trained models.

We recognize that technologies improving LLM reasoning could be applied for malicious purposes, a risk inherent to progress in this field. Our primary objective is to advance the scientific understanding of efficient, structured reasoning and make powerful AI techniques more accessible and scalable. A significant positive ethical implication of our work is the substantial reduction in computational resources required for complex reasoning tasks. Our method lowers the financial and environmental costs associated with running large models, thereby promoting more equitable access to advanced AI capabilities.

The foundational models (e.g., Qwen3, Llama3.1, Gemma3) and datasets (e.g., GSM8K, GPQA) used in our experiments may contain existing societal biases. Our proposed method, DST, does not explicitly mitigate these biases but rather focuses on the structural efficiency of the reasoning process. The potential for the predictor to inadvertently learn or amplify these biases is a limitation and an important direction for future research. We believe the benefits of enabling more efficient and scalable reasoning outweigh the immediate risks, which are common to most research in this domain.

## REPRODUCIBILITY STATEMENT

To ensure the reproducibility of our findings, we have provided a detailed account of our methodology, experimental setup, and results. The core algorithms for training the DST predictor and performing guided tree search are formally described in Section 3, with specific pseudocode in Algorithm 1 and Algorithm 2. All datasets used in our experiments—including GSM8K, SVAMP, MATH-500, GPQA, and subsets of BBEH—are standard public benchmarks; further details are available in Appendix A.

Our experimental setup, including the specific backbone models (Qwen3-8B, Llama3.1-8B-Instruct, Gemma3-12B-it), baselines, and evaluation metrics, is described in Section 4.1. Comprehensive details regarding hyperparameters and hardware configurations are provided in Appendix B. We include extensive ablation studies in Section 4.2 to analyze the impact of individual model components and key hyperparameters such as beam width, pruning threshold, and the discount factor, with results visualized in Figures 2, 3, and 4. To facilitate direct replication and further research, we make our source code, including scripts for predictor training, data generation, and evaluation, publicly available upon acceptance at https://anonymous.4open.science/r/CoTPruning-2308.

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

## A  DATASETS AND BASELINES

We utilized a diverse set of standard public benchmarks to rigorously evaluate the performance of our method across different reasoning domains. The detailed information is as follows.

- **GSM8K** is a famous dataset containing 1319 primary school level math problems. It is widely recognized as a standard for gauging fundamental quantitative reasoning and the ability to translate natural language descriptions into mathematical operations. It serves as a baseline for core numerical and logical abilities.

- **MATH-500** is a curated and high-quality subset of 500 challenging problems extracted from the comprehensive MATH test set . Sourced from American high school mathematics competitions, this dataset covers seven distinct subjects, including algebra, geometry, number theory, and precalculus, thereby demanding more sophisticated problem-solving heuristics than simple arithmetic .

- **Minerva-Math** is a specialized collection designed for training and evaluating AI models on challenging mathematical reasoning tasks. It includes 272 problems ranging from algebra and calculus to advanced proofs, testing the model's ability to engage with more abstract and rigorous mathematical thought processes.

- **SVAMP** is a dataset containing 1,000 math word problems designed to test a model's robustness to linguistic variations. By systematically modifying existing problems from other datasets, SVAMP assesses whether a model's reasoning abilities are brittle and overly sensitive to minor changes in sentence structure and question phrasing for one and two-step arithmetic problems.

- **GPQA** is a challenging dataset of 448 graduate-level, multiple-choice questions in biology, physics, and chemistry, authored by domain experts. The questions are designed to be "Google-proof", meaning they are difficult for non-experts to answer even with access to a search engine, thus rigorously testing the expert-level knowledge and reasoning capabilities of advanced AI systems.

- **Big-bench Extra Hard** is a challenging subset of the BIG-Bench suite, consisting of 23 tasks that were identified as being particularly difficult for contemporary language models at the time of its release. These tasks are diverse and complex, including causal judgment, formal fallacies, logical deduction, tracking shuffled objects, and navigating a grid. High performance on this benchmark requires robust multi-step reasoning capabilities and the ability to follow intricate instructions.

Our method was compared against three key baseline approaches to demonstrate its superior accuracy and efficiency.

- **Chain-of-Thought** (CoT) is a standard prompting technique that elicits reasoning by instructing the model to generate a series of intermediate steps that lead to a final answer. It is often implemented using few-shot examples and serves as the foundational baseline for reasoning performance.

- **Tree-of-Thoughts** (ToT) is a framework that models problem-solving as a tree search, allowing the model to explore multiple reasoning paths concurrently. The standard implementation uses an expensive, LLM-based self-evaluation mechanism to score and prune branches, representing a powerful but computationally intensive upper baseline.

- **Dynamic Parallel Tree Search** (DPTS) is a recent adaptive variant of ToT that uses local confidence scores derived from the model's own logits to guide a parallel, breadth-first expansion. It aims to improve efficiency over the standard ToT by avoiding explicit LLM-based evaluators but can be limited by the reliability of confidence scores as a predictor of future success.

## B  EXPERIMENT DETAILS

This section outlines the specific configurations and hyperparameters used in our experiments to ensure reproducibility.

**Backbone Models.** All experiments were conducted using the following publicly available backbone language models.

- **Qwen3-8B** is an 8-billion parameter model from the Qwen3 series developed by Alibaba Cloud.
- **Llama3.1-8B-Instruct** is an 8-billion parameter, instruction-tuned model from the Llama 3.1 family developed by Meta.
- **Gemma3-12B-it** is a 12-billion parameter, instruction-tuned model from the Gemma 3 family developed by Google.

**Hardware Configuration.** To ensure a fair comparison across all methods and models, experiments were performed on an identical hardware setup. We conduct our experiments on a server with 64 cores Intel Xeon 2.90GHz CPU, 256 GB RAM, and 4 NVIDIA 3090 GPUs running the Ubuntu 20.04 operating system.

**Hyperparameter Settings.** Consistent hyperparameters were used for all main experiments unless otherwise specified in the ablation studies.

**DST Predictor Training.** The DST predictor was implemented using a LightGBM classifier, a highly efficient gradient boosting framework. This model was trained on features extracted from successful and unsuccessful reasoning traces to learn how to distinguish between promising and unpromising solution paths. Key hyperparameters for training included a learning rate of 0.05, 500 boosting estimators, and a maximum of 31 leaves per tree to control model complexity and prevent overfitting on the training data.

**Runtime Inference and Generation.**

- Beam Width $b$. The default maximum beam width was set to 3, as this value was found to offer a strong balance between performance and computational cost (see Figure 3).
- Pruning Threshold $\tau$: The default pruning threshold was set to 0.7 for Math and GPQA and 0.8 for BBEH subtasks, based on the saturation point observed in our sensitivity analysis (see Figure 4).
- Discount Factor $\gamma$: The default score propagation discount factor was set to 0.99, which empirically yielded the highest accuracy by balancing solution brevity and completeness (see Figure 5).
- Temperature: A temperature setting of 0.7 was used for LLM generation across all experiments. This non-zero value encourages the generation of diverse candidate thoughts at each step of the tree search, which is essential for effective exploration.

## C  SENSITIVITY TO HYPERPARAMETERS

This section analyzes the model's sensitivity to three critical hyperparameters: the beam width $b$, the pruning threshold $\tau$, and the score propagation discount factor $\gamma$.

**Beam width $b$.** To analyze the effect of exploration on solution quality, we vary the maximum beam width $b$ on the BBEH-BoardgameQA dataset. The results, shown in Figure 3, illustrate the fundamental trade-off between the breadth of the search and the computational resources required.

The figure yields several key insights. First, the most significant performance gain occurs when moving from a narrow beam to a moderate one. Increasing the beam width from $b = 1$ (34.0% accuracy) to $b = 3$ (46.0% accuracy) provides a substantial 12-point absolute improvement. This sharp increase underscores the critical importance of exploring multiple reasoning paths. A purely greedy approach ($b = 1$) is highly susceptible to early-stage errors, and even a modest increase in exploration breadth allows the model to circumvent these pitfalls and find more robust solutions. Second, Beyond $b = 3$, the accuracy curve flattens significantly, demonstrating a clear pattern of diminishing returns. The accuracy gain from $b = 3$ to $b = 5$ is only 0.8 points, and the gain from

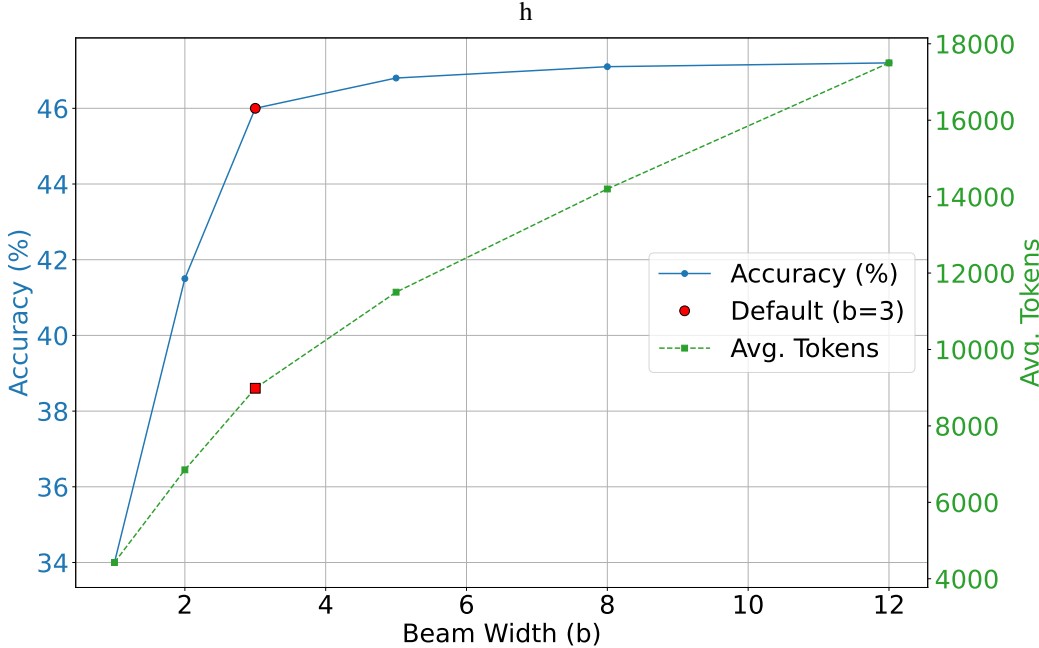

Figure 3: Accuracy vs. Average Tokens as a function of Beam Width ($b$) on BBEH-BoardgameQA. The red dot marks our default setting of $b = 3$, which offers a strong balance between performance and cost.

$b = 5$ to $b = 12$ is less than 0.5 points combined. In stark contrast, the average token consumption (green dashed line) continues to increase in a near-linear fashion. This indicates that while some exploration is crucial, an excessively wide beam provides minimal additional benefit and incurs a prohibitive computational cost, likely due to the inherent reasoning limitations of the backbone LLM.

This analysis confirms that a fixed, wide beam is computationally inefficient. It motivates our core contribution: an adaptive search mechanism that can dynamically prune the search space, aiming to achieve the accuracy of a wide-beam search with the efficiency of a much narrower one.

**Pruning threshold $\tau$.**  The plots for both GSM8K and BBEH-BoardgameQA (top row of Figure 4) demonstrate the fundamental trade-off governed by $\tau$. As $\tau$ increases from 0.5 to 0.95, we consistently observe that accuracy improves while the average token consumption also rises. This is because a higher threshold $\tau$ imposes a stricter confidence requirement for taking a greedy short-cut, forcing the system to default more frequently to the safer, full-beam exploration mode. This increased exploration allows the model to recover from potential early-stage errors and discover higher-quality reasoning paths, thus boosting accuracy at the expense of computational resources. Crucially, both datasets exhibit a plateau effect, where accuracy gains diminish significantly at higher $\tau$ values. For GSM8K, accuracy saturates around $\tau = 0.7$, while for BBEH-BoardgameQA, the curve flattens after $\tau = 0.8$. This indicates that beyond a certain point, the marginal benefit of increased exploration is outweighed by the linear increase in token cost, converging towards performance of a non-adaptive, full beam search.

The "Shortcut Rate Comparison" plot (bottom row of Figure 4) offers direct insight into the adaptive behavior of our predictor. As theoretically predicted, the Shortcut Rate decreases monotonically as $\tau$ increases for both tasks. On GSM8K, where reasoning paths are more uniform, the predictor confidently identifies promising steps and triggers frequent shortcuts. On BBEH-BoardgameQA, the inherent ambiguity of the task leads to lower predictor confidence, resulting in fewer shortcuts and more cautious exploration. The experiments demonstrate that our adaptive pruning strategy, controlled by a single parameter $\tau$, allows practitioners to navigate the accuracy-efficiency Pareto frontier and tailor the reasoning process to specific deployment constraints and task difficulties.

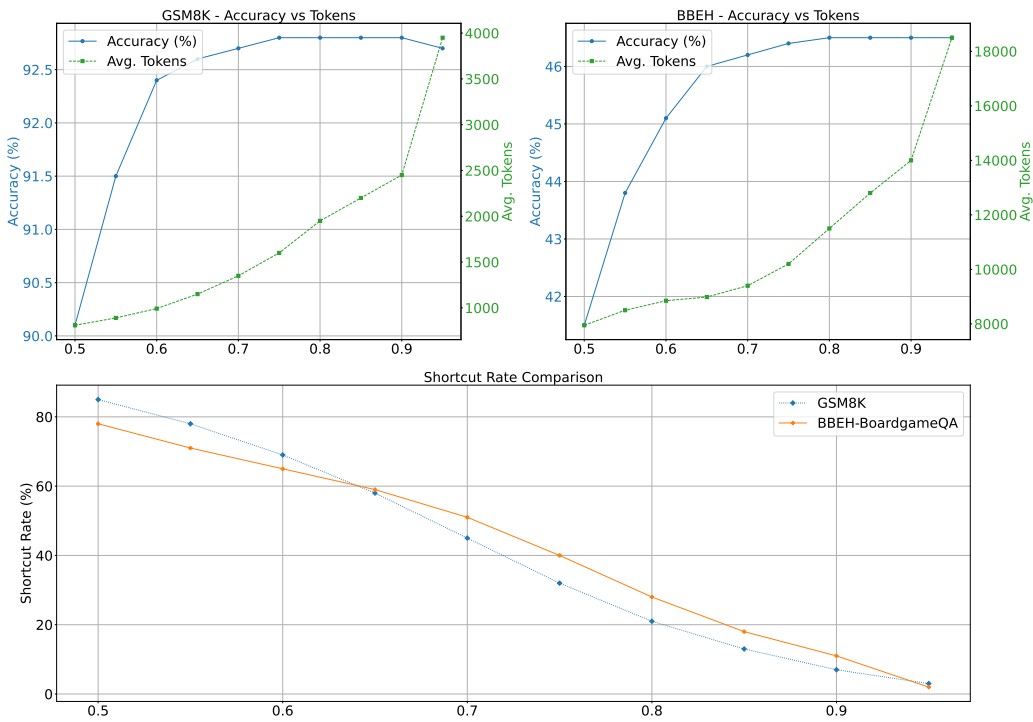

Figure 4: Top: Accuracy vs. Average Tokens as a function of Pruning Threshold ($\tau$) on GSM8K (left) and BBEH-BoardgameQA (right). Bottom: The corresponding Shortcut Rate for each dataset.

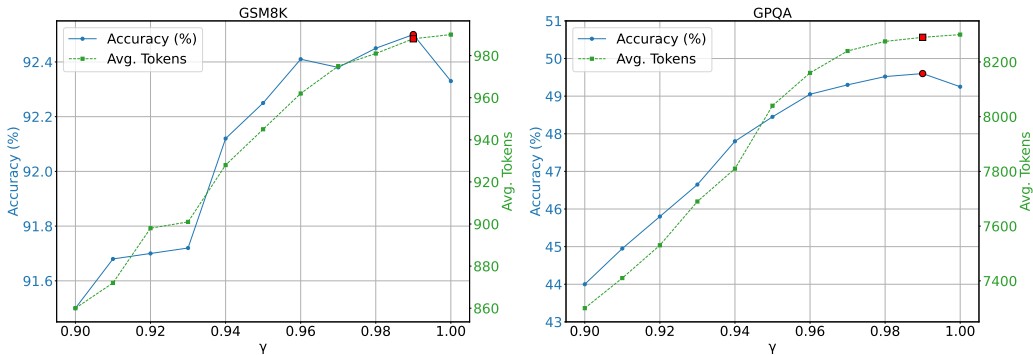

Figure 5: Accuracy vs. Average Tokens as a function of Discount Factor ($\gamma$) on GSM8K (left) and GPQA (right). The red marker indicates the optimal performance point, achieved at $\gamma = 0.99$.

**Discount factor $\gamma$.** To investigate how an inductive bias towards solution conciseness affects reasoning quality, we analyze the performance impact of the discount factor $\gamma$. This hyperparameter, used during the score propagation phase of predictor training, discounts the value of longer reasoning chains. We systematically evaluate $\gamma$ in the range $[0.90, 1.00]$ on both the structured GSM8K dataset and the more complex GPQA dataset.

The results, visualized in Figure 5, reveal a distinct and non-linear relationship, supporting our hypothesis that an optimal balance exists between encouraging brevity and allowing for sufficient reasoning depth. For both GSM8K and GPQA, the maximum accuracy is achieved precisely at $\gamma = 0.99$, which we select as our default setting (indicated by the red marker). The performance drop from $\gamma = 0.99$ to $\gamma = 1.00$ suggests that having no penalty against verbosity is suboptimal. Allowing unconstrained path lengths may lead the model down convoluted or error-prone reasoning

trajectories. The significant accuracy loss at lower $\gamma$ values (e.g., 0.90) confirms that an overly aggressive penalty is also detrimental, as it discourages the model from taking necessary, multi-step reasoning actions, particularly on complex problems like GPQA.

Our empirical results validate that a carefully calibrated penalty against verbosity is superior to both extreme brevity and unconstrained exploration, providing a principled foundation for our training methodology.

## D  THE USE OF LARGE LANGUAGE MODELS

In the preparation of this manuscript, we utilized the Large Language Model (LLM) Gemini 2.5 Pro. The role of the LLM was strictly limited to that of a general-purpose writing assistant. Specifically, it was used for polishing the manuscript to improve grammar, refine phrasing, and enhance the overall clarity and readability of the text. All core scientific contributions, including the research ideation, methodological design, experimental setup, data analysis, and the initial drafting of all content, were performed exclusively by the authors. The authors have carefully reviewed all suggested edits and take full responsibility for the final content of this paper, including its scientific accuracy and integrity.

