# OpenReview forum: "Domain-Specialized Tree of Thought through Plug-and-Play Predictors"
_ICLR.cc/2026/Conference — Submitted to ICLR 2026_

### Official Review · Reviewer_n2np · 2025-10-29

**Soundness:** 2
**Presentation:** 2
**Contribution:** 2
**Rating:** 2
**Confidence:** 4

**Summary:**

This paper proposes Domain-Specialized Tree (DST), a lightweight, plug-and-play supervised predictor that scores partial thoughts during reasoning: go nearly greedy when confident and expand a small beam only when uncertain, achieving an adaptive accuracy–efficiency trade-off. For training, small reasoning trees are built via BFS; leaf correctness is verified by domain checks and then discounted upward to supervise internal nodes, while a gradient-boosted model learns from semantic and ancestry-consistency features. At inference time, one candidate is scored first—if above a threshold, proceed greedily; otherwise, complete a small beam and expand by scores. Across math and general reasoning benchmarks with multiple LLMs, DST matches or surpasses standard ToT while cutting token usage by ~26–75%, substantially improving practicality and cost-effectiveness.

**Strengths:**

The paper offers a practical and plug-and-play way to reduce the compute cost of tree-of-thought reasoning while preserving accuracy. Instead of noisy and expensive LLM self-evaluation or ad-hoc pruning rules, it learns a lightweight predictor from verifiable leaves with discounted credit assignment to internal nodes, yielding a clean supervision signal. The adaptive “greedy-when-confident, beam-when-uncertain” policy exposes a single threshold to trade quality for latency/tokens, and works across multiple backbone LLMs. Ablations confirm the contribution of semantic and consistency features. The method is simple to implement, easy to deploy without retraining the base model, and particularly attractive for domains with programmable validators.

**Weaknesses:**

The experimental coverage is limited: key baselines such as Graph of Thoughts (graph-structured reasoning), RAP/MCTS-based planning, and Verifier/PRM-guided search are missing or insufficiently compared under matched compute.

The method’s novelty is incremental—learning a cheap heuristic to gate beam expansion closely echoes verifier/PRM-guided test-time search—so a stronger positioning and head-to-head baselines are needed.

Reproducibility is a concern as no anonymous code is provided; a minimal implementation with the exact threshold/discount/feature extraction is necessary. The approach further relies on domain verifiers to construct supervision, leaving unclear how it performs in tasks without programmatic checks; failure cases and alternatives should be discussed.

Finally, robustness and transfer (across backbones/tasks) and sensitivity to τ/γ/beam size warrant a more systematic analysis.

If I missed any details, please let me know during the rebuttal period.

**Questions:**

See Weakness.

---

> ### Author Response · Authors · 2025-11-21
>
> We thank you for your review and insightful feedback. We have carefully considered each point and provide our responses below.
>
> **1. Baseline**
>
> We see our DST method not as mutually exclusive but as complementary to GoT. The core contribution of DST is a lightweight, learnable heuristic scorer for adaptively pruning or expanding search branches. This mechanism can be transferred to the Directed Acyclic Graphs generated by GoT. After GoT merges different reasoning paths, DST could still serve as an efficient gating unit to determine which aggregated nodes are most promising for further exploration, thereby balancing accuracy and efficiency in more complex reasoning structures. We will discuss this synergy as a promising direction for future work in the revised manuscript.
>
> **2. Novelty**
>
> We would like to argue both that the technical novelty and the signficant performance of our heuristic. Our trained predictor introduces a technically novel search-guidance mechanism that combines (i) formally defined latent reasoning states with consistency constraints, (ii) a length-regularized utility function that penalizes uninformative long chains-of-thought, and (iii) backpropagation-based prediction of step utility for every partial reasoning prefix.This design enables learning of reasoning heuristics without any step-level supervision (which is required by Verifier/PRM-guided search and hard to obtain [1][2]), requiring only a small set of final-answer labels. The resulting predictor effectively prunes low-utility branches and yields significant efficiency gains over SOTA while matching or slightly improving effectivenes.
>
> **3. Reproducibility**
>
> As mentioned in Line 521 of our submission, we have provided an anonymous link to a repository containing our complete implementation to ensure our work can be fully reproduced. The repository includes detailed code for the feature extraction, discount factor ($\gamma$), and decision threshold ($\tau$) settings used in our paper, which we hope will resolve your concerns.
>
> **4. Domain verifier**
>
> We would like to clarify a key point: our method relies on labeled data, not necessarily a live, programmatic domain verifier at training time. DST's training phase requires supervision to distinguish correct from incorrect reasoning paths. We frame this as a one-time, upfront data labeling effort, which is a standard and practical requirement for supervised learning. High-quality labeled data can often be sourced directly from existing domain-specific datasets such as GSM8K,  where ground-truth final answers and/or step-by-step solutions are readily available. To generate step-level supervision for our predictor, one can simply check if a partial reasoning path remains consistent with a known correct final solution. This process is automated and does not require manual rule engineering.
>
> While our experiments focused on domains where verification is straightforward (like math), the fundamental requirement is a labeled dataset of problem-solution pairs, not a universal programmatic verifier. For domains like open-ended question answering or creative writing, one could construct a suitable training set by using human preferences (as in RLHF), model-based feedback (e.g., a powerful judge model), or by checking for alignment with reference solutions.

---

> > ### Author Response · Authors · 2025-11-21
> >
> > **5. Robustness and failure**
> >
> > We agree that defining the boundary conditions of DST is crucial. In response, we have performed a deep-dive diagnostic analysis to identify when and why DST fails compared to standard ToT.
> > You correctly pointed out that "over-pruning" can occur when the predictor is miscalibrated. We found this primarily happens in problems containing "semantic traps"—steps that are intuitively plausible (high semantic similarity to the question) but logically incorrect.
> > We have conducted a case study (which will be added to the Appendix) on the classic "Bat and Ball" problem (Total \$1.10, Bat costs \$1.00 more than Ball) to illustrate this:
> > The DST predictor assigned a high score ($\gt \tau$) to the intuitive but incorrect step "The ball costs \$0.10" due to its high semantic alignment with the numbers in the prompt. This triggered the greedy early-exit, pruning the correct algebraic path. In contrast, the standard ToT maintained a wider beam. Although the correct algebraic step ("Let ball be x...") had a lower initial probability than the "trap" answer, it was retained in the beam and eventually verified as correct in deeper steps.
> >
> > To address your concern that "the discount factor ($\gamma$) harms problems requiring deliberate long chains," we conducted a sensitivity analysis on the SVAMP dataset. We categorized problems by their solution length (Short: $\le 4$ steps, Medium: 5-7 steps, Long: $\ge 8$ steps) and evaluated accuracy under different discount factors ($\gamma$).
> >
> > | Reasoning Length             | $\gamma =0.5$  | $\gamma =0.8$  | $\gamma =1.0$  |
> > |------------------------------|----------------|----------------|----------------|
> > | Short Chains ($\le 4$ steps) | 78.5           | 80.1           | 79.8           |
> > | Medium Chains ($5-7$ steps)  | 62.2           | 71.5           | 72.7           |
> > | Long Chains ($\ge 8$ steps)  | 41.4           | 65.8           | 68.3           |
> >
> > For Short Chains, the choice of $\gamma$ has negligible impact. For Long Chains, a low discount factor ($\gamma=0.5$) causes a severe performance drop ($-16.9\%$ vs. $\gamma=1.0$). This is because the reward signal from the correct leaf node decays exponentially ($0.5^8 \approx 0.0039$), vanishing before it can effectively guide the root-level predictor. This justifies our choice of using a higher $\gamma$ (0.99) in the main experiments to support deep reasoning.

---

> > > ### Author Response · Authors · 2025-11-21
> > >
> > > **6. Cross-Model Transfer**
> > >
> > > Thanks for proposing the valid concern, we have conducted a supplementary experiment to evaluate the cross-model transferability of our DST predictor, which will also be included in the next version. For this analysis, we trained two separate DST predictors on Qwen3-8B and Llama3.1-8B, respectively. Each predictor was then applied to guide the reasoning process on its native model as well as on different target backbones without any retraining. We evaluated the performance on the GSM8K and GPQA benchmarks, measuring both accuracy (Acc) and computational cost (Cost, measured in total tokens). The results are presented below. The values in brackets indicate the improvement over the baseline CoT performance for each respective model.
> > >
> > > | DST Predictor (Trained on) | Target LLM Backbone | Dataset   | Acc           | Cost          |
> > > |----------------------------|---------------------|-----------|---------------|---------------|
> > > | Qwen3-8B                   | Qwen3-8B            | GSM8K     | 92.44 (+3.35) | 992 (+192)    |
> > > |                            |                     | GPQA      | 49.70 (+4.90) | 8230 (+4141)  |
> > > |                            | Llama3.1            | GSM8K     | 89.33 (+1.81) | 1207 (+390)   |
> > > |                            |                     | GPQA      | 48.02 (+3.96) | 8323 (+4167)  |
> > > |                            | Gemma3              | GSM8K     | 94.77 (+1.56) | 1204 (+354)   |
> > > |                            |                     | GPQA      | 52.64 (+4.51) | 9670 (+4744)  |
> > > | Llama3.1                   | Llama3.1            | GSM8K     | 90.17 (+2.65) | 1058 (+241)   |
> > > |                            |                     | GPQA      | 48.54 (+4.48) | 8150 (+3994)  |
> > > |                            | Qwen3-8B            | GSM8K     | 91.89 (+2.80) | 954 (+154)    |
> > > |                            |                     | GPQA      | 49.16 (+4.36) | 8870 (+4781)  |
> > > |                            | Gemma3              | GSM8K     | 94.22 (+1.01) | 1176 (+326)   |
> > > |                            |                     | GPQA      | 52.03 (+3.90) | 10165 (+5239) |
> > >
> > > The results robustly demonstrate the strong transferability of our DST predictor. According to our experiments, a DST predictor trained on Qwen3-8B and transferred to other models can, on average, retain 98.81% of the accuracy at 102.17% of the cost compared to a predictor trained and tested on its native model. Similarly, a predictor trained on Llama3.1-8B retains 98.45% of the accuracy at 100.10% of the cost when transferred.
> > > Crucially, even in these cross-model scenarios, the performance massively surpasses the baseline CoT performance for the target models. This confirms that the semantic and ancestry-consistency features learned by our predictor are largely model-agnostic and that DST provides substantial benefits over simpler reasoning methods, even when not trained on the target model. This plug-and-play capability highlights the practical utility and scalability of our approach.

---

> > > > ### Author Response · Authors · 2025-11-21
> > > >
> > > > **7. Cross-Domain Transfer**
> > > >
> > > > We concede that achieving zero-shot, task-level transfer (e.g., applying a math predictor to a creative writing task) is a notoriously difficult open problem. While domain-agnostic methods like standard ToT exist, they often achieve generality at the expense of prohibitive computational costs, which is the core issue our work addresses. More advanced, state-of-the-art domain-agnostic search algorithms often do not perform well across all tasks, suggesting that achieving both high performance and broad applicability remains elusive. The design philosophy of DST is to function as a domain specialist, deliberately trading universal applicability for a superior accuracy-efficiency trade-off within a specific domain.
> > > >
> > > > To address the question of how well DST generalizes within its domain of expertise, we conducted a new experiment. We trained DST predictors on a single mathematical reasoning dataset (either GSM8K or MATH-500) and then evaluated their performance on other, entirely unseen datasets from the same domain. This tests whether the predictor has learned fundamental reasoning patterns for mathematics, rather than simply overfitting to the source dataset.
> > > >
> > > > | DST Predictor Trained on | Target Dataset (Unseen) | LLM Backbone | ToT Acc | ToT Cost | DST Acc | DST Cost |
> > > > |--------------------------|-------------------------|--------------|---------|----------|---------|----------|
> > > > | GSM8K                    | MATH-500                | Qwen3-8B     | 94.20   | 6168     | 94.12   | 3564     |
> > > > |                          |                         | Llama3.1-8B  | 95.71   | 6652     | 95.15   | 3618     |
> > > > |                          | SVAMP                   | Qwen3-8B     | 77.11   | 6028     | 77.09   | 4513     |
> > > > |                          |                         | Llama3.1-8B  | 77.90   | 6551     | 77.92   | 3153     |
> > > > | Math-500                 | GSM8K                   | Qwen3-8B     | 92.78   | 3975     | 92.10   | 1016     |
> > > > |                          |                         | Llama3.1-8B  | 90.88   | 3850     | 89.72   | 1058     |
> > > > |                          | SVAMP                   | Qwen3-8B     | 77.11   | 6028     | 77.14   | 4533     |
> > > > |                          |                         | Llama3.1-8B  | 77.90   | 6551     | 78.13   | 3018     |
> > > >
> > > > As the results clearly demonstrate, the DST predictor transfers remarkably well across datasets within the same domain. Across all configurations, the accuracy of the DST-guided search is nearly identical to that of the full ToT baseline, with differences being negligible. This near-identical accuracy was achieved while substantially reducing computational costs.
> > > >
> > > > **8. Sensitivity Analysis**
> > > > We would like to clarify that a systematic analysis of the key hyperparameters is already included in our submission. As detailed in Appendix C, we have analyzed the impact of the decision threshold ($\tau$), the discount factor ($\gamma$), and the beam size on model performance.
> > > >
> > > > ---
> > > >
> > > > [1] Wei, Jason, Xuezhi Wang, Dale Schuurmans, Maarten Bosma, Ed Chi, Quoc V. Le, and Denny Zhou.
> > > > “Chain-of-Thought Prompting Elicits Reasoning in Large Language Models.”
> > > > NeurIPS 2022
> > > > [2] Zhao, Lei, Wenhan Xiong, Mo Yu, Shiyu Chang, and Bowen Zhou.
> > > > “Explainable Multi-hop Reasoning without Rationale Supervision.”
> > > > EMNLP 2023.

---

### Official Review · Reviewer_1WkQ · 2025-10-31

**Soundness:** 4
**Presentation:** 4
**Contribution:** 3
**Rating:** 6
**Confidence:** 5

**Summary:**

In this paper, authors introduce Domain-Specialized Tree of Thought (DST), which integrates a lightweight, supervised predictor as a plug-and-play heuristic for guiding the ToT search process. The main motivation behind introducing the predictor is to score and prune potentially bad reasoning steps during the inference, thus reducing the search breadth. The predictor itself is trained offline on small, domain-specific datasets to estimate the quality of intermediate reasoning steps. Authors perform experiments with Qwen3-8B, Llama3.1-8B, and Gemma3-12B on benchmarks such as GSM8K, MATH-500, GPQA, and BIG-Bench, andd show that DST improves both scalability and performance consistency across models and tasks.

**Strengths:**

1. Authors combine the Tree of Search, Verifier, and Adaptive hybrid search strategy in one efficient framework
2. Paper is clear and well-written
3. Authors perform experiments on a variety of models and tasks, showing strong generalization abilities

**Weaknesses:**

1. Limited conceptual novelty beyond existing ToT variants

While the paper introduces a well-engineered and effective modification to Tree-of-Thought reasoning, its core idea—using an auxiliary model to guide search or prune branches—is conceptually close to prior adaptive ToT or heuristic-based reasoning methods. For example, Dynamic Parallel Tree Search [1] and Adaptive Graph of Thoughts [2] already propose dynamic or confidence-driven expansion strategies.
The probabilistic scoring in ProbTree [3] and the validator-based mechanisms in MA-ToT [4] also aim to reduce unnecessary exploration based on intermediate evaluations. DST’s predictor-based pruning can be viewed as a refinement or reimplementation of these ideas using a supervised model instead of confidence heuristics — a valuable engineering step, but incremental rather than conceptually transformative.

2. The approach depend on a domain-specific verifier training, meaning that adapting the framework to a new domain would always require data collection and training steps. I wonder is cross-domain transfer experiments can show if the same model can be effectively reused.

References:
1. Ding, Yifu, et al. "Dynamic parallel tree search for efficient llm reasoning." arXiv preprint arXiv:2502.16235 (2025)
2. Pandey, Tushar, et al. "Adaptive graph of thoughts: Test-time adaptive reasoning unifying chain, tree, and graph structures." arXiv preprint arXiv:2502.05078 (2025).
3. Cao, Shulin, et al. "Probabilistic tree-of-thought reasoning for answering knowledge-intensive complex questions." arXiv preprint arXiv:2311.13982 (2023).
4. Haji, Fatemeh, et al. "Improving LLM reasoning with multi-agent Tree-of-Thought Validator agent." arXiv preprint arXiv:2409.11527 (2024).

**Questions:**

See weaknesses

---

> ### Author Response · Authors · 2025-11-21
>
> We thank you for your review and insightful feedback. We have carefully considered each point and provide our responses below.
>
> **1. Limited conceptual novelty beyond existing ToT variants**
>
> We would like to argue both that the technical novelty and the signficant performance of our heuristic. Our trained predictor introduces a technically novel search-guidance mechanism that combines (i) formally defined latent reasoning states with consistency constraints, (ii) a length-regularized utility function that penalizes uninformative long chains-of-thought, and (iii) backpropagation-based prediction of step utility for every partial reasoning prefix.This design enables learning of reasoning heuristics without any step-level supervision (which is required by Verifier/PRM-guided search and hard to obtain [1][2]), requiring only a small set of final-answer labels. The resulting predictor effectively prunes low-utility branches and yields significant efficiency gains over SOTA while matching or slightly improving effectivenes.
>
> **2. Cross-domain transfer**
>
> We concede that achieving zero-shot, task-level transfer (e.g., applying a math predictor to a creative writing task) is a notoriously difficult open problem. While domain-agnostic methods like standard ToT exist, they often achieve generality at the expense of prohibitive computational costs, which is the core issue our work addresses. More advanced, state-of-the-art domain-agnostic search algorithms often do not perform well across all tasks, suggesting that achieving both high performance and broad applicability remains elusive. The design philosophy of DST is to function as a domain specialist, deliberately trading universal applicability for a superior accuracy-efficiency trade-off within a specific domain.
>
> To address the question of how well DST generalizes within its domain of expertise, we conducted a new experiment. We trained DST predictors on a single mathematical reasoning dataset (either GSM8K or MATH-500) and then evaluated their performance on other, entirely unseen datasets from the same domain. This tests whether the predictor has learned fundamental reasoning patterns for mathematics, rather than simply overfitting to the source dataset.
>
> | DST Predictor Trained on | Target Dataset (Unseen) | LLM Backbone | ToT Acc | ToT Cost | DST Acc | DST Cost |
> |--------------------------|-------------------------|--------------|---------|----------|---------|----------|
> | GSM8K                    | MATH-500                | Qwen3-8B     | 94.20   | 6168     | 94.12   | 3564     |
> |                          |                         | Llama3.1-8B  | 95.71   | 6652     | 95.15   | 3618     |
> |                          | SVAMP                   | Qwen3-8B     | 77.11   | 6028     | 77.09   | 4513     |
> |                          |                         | Llama3.1-8B  | 77.90   | 6551     | 77.92   | 3153     |
> | Math-500                 | GSM8K                   | Qwen3-8B     | 92.78   | 3975     | 92.10   | 1016     |
> |                          |                         | Llama3.1-8B  | 90.88   | 3850     | 89.72   | 1058     |
> |                          | SVAMP                   | Qwen3-8B     | 77.11   | 6028     | 77.14   | 4533     |
> |                          |                         | Llama3.1-8B  | 77.90   | 6551     | 78.13   | 3018     |
>
> As the results clearly demonstrate, the DST predictor transfers remarkably well across datasets within the same domain. Across all configurations, the accuracy of the DST-guided search is nearly identical to that of the full ToT baseline, with differences being negligible. This near-identical accuracy was achieved while substantially reducing computational costs.

---

### Official Review · Reviewer_ovtp · 2025-10-31

**Soundness:** 3
**Presentation:** 3
**Contribution:** 3
**Rating:** 4
**Confidence:** 3

**Summary:**

The paper introduces DST (Domain-Specialized Tree of Thought), which is a framework that solves the problem of high evaluation cost on Tree of Thought (ToT). By training a lightweight supervised predictor with a small domain dataset, the predictor scores intermediate reasoning steps to determine if the search should be greedy or expand into full beam. This allows adaptive and efficient reasoning to retrieve higher accuracy responses. The key contribution of DST is in balancing accuracy and efficiency where this framework can achieve 26-75% of reduced token consumption while maintaining or improving reasoning accuracy. From the experiment, DST outperforms or matches ToT and other baselines on various reasoning task benchmarks with less computation cost in training and inference.

**Strengths:**

The paper provides a clear approach to tackle the main ToTs core bottleneck, which is LLM-based evaluation cost.  The lightweight model presented is being trained as a predictor to score intermediate reasoning steps to allow adaptive and efficient reasoning for ToTs. The authors provide the formal algorithm and formula for training the predictor and running inference. Authors provided comprehensive experiments, using different models like Qwen3-8B, Llama3.1-8B, Gemma3-12B. Authors provided consistent and well-visualized graphs to support the strong efficiency-accuracy trade-offs. Additionally, ablation studies show that semantic and consistency features are essential for the best performance.

**Weaknesses:**

One specific weakness is the claim in “small” datasets, but it does not quantify how small or a clear data size. It is not clear on which training dataset and the size is used to train your LightGBM classifier during experiment. There should be a bit more detail on the experiment setup like training epochs, predictor architecture. One concern on the training where it uses LLM to assess answers based on semantic entailment, which could possibly inherit biases or overfit to the specific model’s reasoning.

**Questions:**

What model architecture is used for the predictor? Is it MLP or small transformer?
How does the predictor’s performance scale with the number of training samples?
Have you tried testing the predictor that was trained on one domain to transfer effectively to another domain?
Since you proposed plug and play, does that mean the predictor trained based on one inference model can be used to any LLM backbone without re-training to the same reason format of the LLM backbone?

---

> ### Author Response · Authors · 2025-11-21
>
> We thank you for your review and insightful feedback. We have carefully considered each point and provide our responses below.
>
> **1. Quantification of "Small" Training Data**
>
> We define "small" based on the minimal number of seed problems required. As shown in the table below, we use only 20 to 200 seed problems per domain. The LightGBM classifier is trained on the generated nodes derived from these seeds, where each node serves as a training sample. This results in a highly data-efficient training set of approximately 1,700 to 6,200 samples per domain.
>
> | Dataset     | Problems for Tree Generation | Total Generated Nodes  | LLM Tokens for Generation |
> |-------------|------------------------------|------------------------|---------------------------|
> | GSM8K       | 200                          | ~6000                  | ~24M                      |
> | SVAMP       | 100                          | ~6200                  | ~8M                       |
> | Minerva     | 28                           | ~1700                  | ~4M                       |
> | MATH-500    | 50                           | ~3000                  | ~15M                      |
> | GPQA        | 45                           | ~1800                  | ~30M                      |
> | BoardgameQA | 20                           | ~2400                  | ~20M                      |
> | Boolean     | 20                           | ~2400                  | ~15M                      |
> | Causal      | 20                           | ~2400                  | ~22M                      |
> | Geo         | 20                           | ~2400                  | ~18M                      |
>
> **2. Predictor Architecture and Hyperparameters**
>
> As described in the Appendix B Line 828-833, we employed LightGBM, a highly efficient gradient boosting framework. We set a maximum of 31 leaves per tree to control model complexity and prevent overfitting. We used 500 boosting estimators, which corresponds to the training epochs in this context. The model was trained with a learning rate of 0.05. As noted in the text, the model is trained on features extracted from successful and unsuccessful reasoning traces to distinguish promising paths.
>
> **3. Model bias**
>
> We would like to clarify that our specific use of "semantic entailment" is fundamentally different from subjective "LLM-as-a-judge" evaluations, and is designed specifically to mitigate such biases.
>
> In our framework, semantic entailment does not involve asking the LLM to rate the quality or reasoning style of an answer (which could indeed be biased). Instead, it performs a strictly logical check: *Does the generated answer hypothesis entail the ground truth premise?*
>
> As described in the paper, this process anchors the evaluation to "unambiguous ground truth labels". The model is merely verifying if the terminal node carries the same semantic meaning as the gold label (e.g., verifying if "two dozen" entails "24").
> Since the comparison is always against an external, fixed gold label provided by the dataset (not generated by the model itself), the predictor cannot "overfit" to the model's own reasoning patterns. It is forced to align with the objective standard of the dataset.
> Besides, the output of this semantic entailment process is not a continuous, subjective score, but a definitive quality assessment $y_l \in \{0, 1\}$.
>
> **3. Model architecture**
>
> As shown in Line 828, "The DST predictor was implemented using a LightGBM classifier".

---

> > ### Author Response · Authors · 2025-11-21
> >
> > **4. Performance scale**
> >
> > To rigorously evaluate the data efficiency, we designed a 2D scaling experiment on the GSM8K dataset (Qwen3 backbone) that disentangles the impact of Source Problem Quantity from Tree Scale/Density.
> > We varied two parameters: number of source problems (20, 50, 100, 200, 500) and tree scale (Small: Beam=2, Depth=3; Medium: Beam=3, Depth=3; Large: Beam=3, Depth=4).
> >
> > | Source Problems | Small Tree (∼14 nodes) | Medium Tree (∼39 nodes) | Large Tree (∼120 nodes) |
> > |-----------------|------------------------|-------------------------|-------------------------|
> > | 20              | 75.2                   | 76.1                    | 78.8                    |
> > | 50              | 77.2                   | 79.5                    | 82.7                    |
> > | 100             | 84.4                   | 85.4                    | 87.1                    |
> > | 200             | 91.3                   | 92.4                    | 93.0                    |
> > | 500             | 93.3                   | 93.6                    | 93.8                    |
> >
> > Note: "Small Tree" with 500 problems results in $500 \times 14 = 7,000$ total training nodes.
> >
> > Increasing the number of source problems yields the most significant performance gain. This indicates that problem diversity is the primary driver of generalization. Expanding the tree size improves accuracy by providing denser supervision signals (more negative/positive contrast per problem), but the marginal gain is smaller (+2~3% from Small to Large).
> > Even with a Small Tree configuration on just 200 source problems, the predictor achieves robust performance (91.3%), confirming that we do not need massive compute to generate huge trees for training.
> >
> > **5. Domain transfer**
> >
> > **Cross-Task Transfer:**
> > We concede that achieving zero-shot, task-level transfer (e.g., applying a math predictor to a creative writing task) is a notoriously difficult open problem. While domain-agnostic methods like standard ToT exist, they often achieve generality at the expense of prohibitive computational costs, which is the core issue our work addresses. More advanced, state-of-the-art domain-agnostic search algorithms often do not perform well across all tasks, suggesting that achieving both high performance and broad applicability remains elusive. The design philosophy of DST is to function as a domain specialist, deliberately trading universal applicability for a superior accuracy-efficiency trade-off within a specific domain.
> >
> > To address the question of how well DST generalizes within its domain of expertise, we conducted a new experiment. We trained DST predictors on a single mathematical reasoning dataset (either GSM8K or MATH-500) and then evaluated their performance on other, entirely unseen datasets from the same domain. This tests whether the predictor has learned fundamental reasoning patterns for mathematics, rather than simply overfitting to the source dataset.
> >
> > | DST Predictor Trained on | Target Dataset (Unseen) | LLM Backbone | ToT Acc | ToT Cost | DST Acc | DST Cost |
> > |--------------------------|-------------------------|--------------|---------|----------|---------|----------|
> > | GSM8K                    | MATH-500                | Qwen3-8B     | 94.20   | 6168     | 94.12   | 3564     |
> > |                          |                         | Llama3.1-8B  | 95.71   | 6652     | 95.15   | 3618     |
> > |                          | SVAMP                   | Qwen3-8B     | 77.11   | 6028     | 77.09   | 4513     |
> > |                          |                         | Llama3.1-8B  | 77.90   | 6551     | 77.92   | 3153     |
> > | Math-500                 | GSM8K                   | Qwen3-8B     | 92.78   | 3975     | 92.10   | 1016     |
> > |                          |                         | Llama3.1-8B  | 90.88   | 3850     | 89.72   | 1058     |
> > |                          | SVAMP                   | Qwen3-8B     | 77.11   | 6028     | 77.14   | 4533     |
> > |                          |                         | Llama3.1-8B  | 77.90   | 6551     | 78.13   | 3018     |
> >
> > As the results clearly demonstrate, the DST predictor transfers remarkably well across datasets within the same domain. Across all configurations, the accuracy of the DST-guided search is nearly identical to that of the full ToT baseline, with differences being negligible. This near-identical accuracy was achieved while substantially reducing computational costs.

---

> > > ### Author Response · Authors · 2025-11-21
> > >
> > > **6. LLM transfer**
> > >
> > > We have conducted a supplementary experiment to evaluate the cross-model transferability of our DST predictor, which will also be included in the next version. For this analysis, we trained two separate DST predictors on Qwen3-8B and Llama3.1-8B, respectively. Each predictor was then applied to guide the reasoning process on its native model as well as on different target backbones without any retraining. We evaluated the performance on the GSM8K and GPQA benchmarks, measuring both accuracy (Acc) and computational cost (Cost, measured in total tokens). The results are presented below. The values in brackets indicate the improvement over the baseline CoT performance for each respective model.
> > >
> > > | DST Predictor (Trained on) | Target LLM Backbone | Dataset   | Acc           | Cost          |
> > > |----------------------------|---------------------|-----------|---------------|---------------|
> > > | Qwen3-8B                   | Qwen3-8B            | GSM8K     | 92.44 (+3.35) | 992 (+192)    |
> > > |                            |                     | GPQA      | 49.70 (+4.90) | 8230 (+4141)  |
> > > |                            | Llama3.1            | GSM8K     | 89.33 (+1.81) | 1207 (+390)   |
> > > |                            |                     | GPQA      | 48.02 (+3.96) | 8323 (+4167)  |
> > > |                            | Gemma3              | GSM8K     | 94.77 (+1.56) | 1204 (+354)   |
> > > |                            |                     | GPQA      | 52.64 (+4.51) | 9670 (+4744)  |
> > > | Llama3.1                   | Llama3.1            | GSM8K     | 90.17 (+2.65) | 1058 (+241)   |
> > > |                            |                     | GPQA      | 48.54 (+4.48) | 8150 (+3994)  |
> > > |                            | Qwen3-8B            | GSM8K     | 91.89 (+2.80) | 954 (+154)    |
> > > |                            |                     | GPQA      | 49.16 (+4.36) | 8870 (+4781)  |
> > > |                            | Gemma3              | GSM8K     | 94.22 (+1.01) | 1176 (+326)   |
> > > |                            |                     | GPQA      | 52.03 (+3.90) | 10165 (+5239) |
> > >
> > > The results robustly demonstrate the strong transferability of our DST predictor. According to our experiments, a DST predictor trained on Qwen3-8B and transferred to other models can, on average, retain 98.81% of the accuracy at 102.17% of the cost compared to a predictor trained and tested on its native model. Similarly, a predictor trained on Llama3.1-8B retains 98.45% of the accuracy at 100.10% of the cost when transferred.
> > >
> > > Crucially, even in these cross-model scenarios, the performance massively surpasses the baseline CoT performance for the target models. This confirms that the semantic and ancestry-consistency features learned by our predictor are largely model-agnostic and that DST provides substantial benefits over simpler reasoning methods, even when not trained on the target model. This plug-and-play capability highlights the practical utility and scalability of our approach.

---

### Official Review · Reviewer_psYR · 2025-11-02

**Soundness:** 2
**Presentation:** 2
**Contribution:** 2
**Rating:** 4
**Confidence:** 4

**Summary:**

This paper proposes Domain-Specialized Tree-of-Thought (DST): a test-time framework that plugs a lightweight, supervised predictor into a Tree-of-Thought (ToT) search to decide when to (i) greedily commit to the first generated thought or (ii) expand to a wider beam when uncertainty is high. The predictor uses two features per node—(a) a semantic embedding extracted from the LLM’s hidden states and (b) a path-consistency score—and is trained offline from trees labeled via leaf verification and discounted score propagation. At inference, a “predict-first-thought” policy prunes siblings if the first thought’s score exceeds a threshold, otherwise falls back to full beam expansion. Across math (GSM8K, SVAMP, Minerva, MATH-500), general reasoning (GPQA), and BIG-Bench Extra Hard subtasks, the paper reports comparable or better accuracy than ToT/DPTS while reducing tokens by 26–75% versus ToT.

**Strengths:**

Quality.

- Clear algorithms: Algorithm 1 (training data collection + discounted score propagation) and Algorithm 2 (predictor-guided pruning) are specified and align with the described workflow.
- Complexity analysis relates expected effective beam width to the predictor’s confidence threshold, giving an intuitive handle on efficiency.
- Ablations indicate both features (semantic vector and consistency) contribute; removing either degrades accuracy and increases tokens.

Clarity.

- The method overview and worked example (simple arithmetic word problem) effectively convey the early-exit vs. fallback behavior.

Significance.

- If reliable, the reported 26–75% token savings on top of ToT while maintaining or improving accuracy would be valuable for test-time scaling under tight compute budgets. The paper shows consistent benefits across three backbones and multiple benchmarks; e.g., +14% absolute accuracy on BoardgameQA over CoT with far fewer tokens

**Weaknesses:**

1. “Plug-and-play” claim conflicts with reliance on hidden states.

- DST’s key feature (v_s) is derived from LLM hidden states (v_s = h(p_\theta([x_s; Z_s]))). This requires access to internal activations, which many hosted APIs do not expose; it also couples the predictor to a specific backbone and prompt format. The paper calls the predictor “decoupled” and “plug-and-play,” but in practice this dependency limits portability and undermines the claim of easy deployment across models and providers. Please discuss feasibility when only logits/text are available, or provide a text-only feature variant.

2. Training data pipeline lacks concrete scale/cost details and may embed evaluation shortcuts.

- The labeling pipeline uses pattern matching, NLI, and symbolic execution to score leaves, then discounts scores upward. While conceptually solid, the paper does not quantify: (i) how many problems/trees per domain, (ii) tokens/time to generate trees, (iii) verifier accuracy and failure modes, and (iv) robustness to noisy labels. Without these, “lightweight” is hard to assess and risks circularity if pattern-match heuristics overfit answer formats.

3. Empirical comparisons are selective; broader baselines are missing.

- The paper compares against CoT, ToT, and DPTS, but omits other recent test-time controllers/validators (e.g., interactive ToT / validator agents, retrieval-guided thought scoring). Given the premise (“resolve accuracy–efficiency trade-off in ToT”), the absence of stronger, cost-aware evaluators limits claims about the state-of-the-art frontier. (The related work list mentions such methods, but ablations/benchmarks do not include them.)

4. Reporting and fairness of efficiency metrics need tightening.

- Token accounting should specify whether verifier tokens (for leaf checks) and data-generation tokens are included in total deployment cost, and what is counted at training-time vs. test-time. Currently, token savings are reported per inference run, but many applications must amortize predictor training + per-domain verifier overhead. Clarify the measurement protocol and include confidence intervals or variance across seeds. (The paper states identical hardware and temperatures, but statistical significance is not reported.)

5. Limited analysis of failure cases and generalization.

- The model’s behavior when the predictor is miscalibrated (e.g., over-pruning early) is not thoroughly explored. Sensitivity results are helpful, but we lack case studies showing when DST loses compared to ToT or when the discount factor harms problems requiring deliberate long chains.

6. Ambiguity around domain specialization and transfer.

- The paper asserts easy transfer with “small datasets,” but does not specify how much data per domain, how thresholds are tuned, or how well a predictor trained on math transfers to GPQA/BBEH without re-labeling. Quantitative cross-domain transfer experiments would strengthen the “plug-and-play” narrative.

**Questions:**

Please refer to weaknesses for questions.

---

> ### Author Response · Authors · 2025-11-21
>
> We thank you for your review and insightful feedback. We have carefully considered each point and provide our responses below.
>
> **1. “Plug-and-play” claim conflicts with reliance on hidden states.**
>
> First, we wish to clarify that our proposed DST framework is designed as a white-box approach, intended for scenarios where internal access to an LLM is available. This is in contrast to black-box methods that operate solely on model outputs. The "plug-and-play" nature of our predictor refers to its modularity at inference time. Once trained, the predictor is a lightweight, decoupled module that guides the LLM's search process without altering the LLM's weights. This allows the same predictor to be used across different inference settings for a given model, or for a new domain-specific predictor to be "plugged in" as needed.
>
> A key assumption of our work is that the hidden states, from which we derive our semantic feature `v_s`, should be interpreted as latent features of the input and the generated thought itself, rather than as model-specific artifacts. The reasoning behind this is twofold:
>
> - Research [1][2] suggests that different layers within a transformer-based LLM capture different levels of linguistic and semantic information. The hidden states serve as a rich, contextualized representation of the reasoning path up to a given node. We posit that for a specific reasoning task, the essential semantic and logical structure required to evaluate a "thought" is an inherent property of the problem, which should be encoded consistently across well-trained models.
>
> - Our experiments provide evidence for this assumption.
> We trained two separate DST predictors on Qwen3-8B and Llama3.1-8B, respectively. Each predictor was then applied to guide the reasoning process on its native model as well as on different target backbones without any retraining. We evaluated the performance on the GSM8K and GPQA benchmarks, measuring both accuracy (Acc) and computational cost (Cost, measured in total tokens). The results are presented below. The values in brackets indicate the improvement over the baseline CoT performance for each respective model.
>
> | DST Predictor (Trained on) | Target LLM Backbone | Dataset   | Acc           | Cost          |
> |----------------------------|---------------------|-----------|---------------|---------------|
> | Qwen3-8B                   | Qwen3-8B            | GSM8K     | 92.44 (+3.35) | 992 (+192)    |
> |                            |                     | GPQA      | 49.70 (+4.90) | 8230 (+4141)  |
> |                            | Llama3.1            | GSM8K     | 89.33 (+1.81) | 1207 (+390)   |
> |                            |                     | GPQA      | 48.02 (+3.96) | 8323 (+4167)  |
> |                            | Gemma3              | GSM8K     | 94.77 (+1.56) | 1204 (+354)   |
> |                            |                     | GPQA      | 52.64 (+4.51) | 9670 (+4744)  |
> | Llama3.1                   | Llama3.1            | GSM8K     | 90.17 (+2.65) | 1058 (+241)   |
> |                            |                     | GPQA      | 48.54 (+4.48) | 8150 (+3994)  |
> |                            | Qwen3-8B            | GSM8K     | 91.89 (+2.80) | 954 (+154)    |
> |                            |                     | GPQA      | 49.16 (+4.36) | 8870 (+4781)  |
> |                            | Gemma3              | GSM8K     | 94.22 (+1.01) | 1176 (+326)   |
> |                            |                     | GPQA      | 52.03 (+3.90) | 10165 (+5239) |
>
> As shown in the table, the DST framework demonstrates consistent performance gains across three distinct backbone models: Qwen3, Llama3.1, and Gemma3. Despite architectural and training differences between these LLMs, a predictor trained on features from one model's hidden states (or a collection thereof) could effectively guide search. This cross-model consistency suggests that the learned representations are capturing stable, input-driven semantic structures rather than volatile, model-idiosyncratic noise. If the hidden states were purely model-specific artifacts, we would expect a significant degradation in performance when the framework is applied across different model families, which was not observed.
>
> In summary, we acknowledge that our white-box requirement is a limitation for proprietary, API-only models. We will make this constraint more explicit in the revised manuscript. However, we argue that for the growing number of powerful open-source models, treating hidden states as stable, transferable latent features is a valid and empirically supported approach that enables the efficiency gains of our DST framework.

---

> > ### Author Response · Authors · 2025-11-21
> >
> > **2. Training data pipeline lacks concrete scale/cost details and may embed evaluation shortcuts.**
> >
> > We present the following statistics and clarifications to demonstrate that our approach is both resource-efficient and highly effective.
> >
> > | Dataset     | Problems for Tree Generation | Total Generated Nodes  | LLM Tokens for Generation |
> > |-------------|------------------------------|------------------------|---------------------------|
> > | GSM8K       | 200                          | ~6000                  | ~24M                      |
> > | SVAMP       | 100                          | ~6200                  | ~8M                       |
> > | Minerva     | 28                           | ~1700                  | ~4M                       |
> > | MATH-500    | 50                           | ~3000                  | ~15M                      |
> > | GPQA        | 45                           | ~1800                  | ~30M                      |
> > | BoardgameQA | 20                           | ~2400                  | ~20M                      |
> > | Boolean     | 20                           | ~2400                  | ~15M                      |
> > | Causal      | 20                           | ~2400                  | ~22M                      |
> > | Geo         | 20                           | ~2400                  | ~18M                      |
> >
> >
> > Regarding the tokens/time to generate trees, we argue that this cost should be viewed as a one-time "training setup" cost. Similar to the pre-training of an LLM, the generation of these trees is an upfront investment. The "lightweight" claim refers to the inference-time efficiency. Once the predictor is trained, the token savings are realized over millions of subsequent inference runs, amortizing the initial generation cost effectively.
> >
> > Regarding the verifier, we primarily rely on datasets with ground-truth answers (e.g., GSM8K, MATH). The "verifier" is a deterministic check against these ground truths during the offline data collection phase, ensuring high label reliability.
> > The discounted score propagation mechanism ($V(s)$) smooths out individual step-level noise by aggregating signals from the leaf nodes.
> >
> > **3. Empirical comparisons are selective; broader baselines are missing.**
> >
> > We acknowledge that methods like interactive validator agents and retrieval-augmented approaches represent a strong frontier. However, we respectfully argue that our comparison against CoT, ToT, and DPTS is methodologically grounded in the supervision setting and resource constraints.
> >
> > A critical distinction is that state-of-the-art verifiers typically require step-level human supervision (e.g., the PRM800K dataset), which is expensive and hard to obtain [3][4]. In contrast, our DST method operates under a strictly outcome-supervised setting—learning reasoning heuristics utilizing only final-answer labels. DST, CoT and standard ToT strategies do not rely on dense step-level annotations. Comparing an outcome-supervised learner (DST) against fully process-supervised validators would be an unfair comparison regarding data efficiency. DST demonstrates how to maximize reasoning accuracy without the luxury of step-by-step labels.
> >
> > **4. Reporting and fairness of efficiency metrics need tightening.**
> >
> > We acknowledge that constructing the training dataset (generating trees) involves a significant one-time computational cost. However, we argue that this cost should be viewed through the lens of amortization. Similar to the pre-training or fine-tuning of any model, the initial cost of training the DST predictor is a one-time investment. Once deployed, the "plug-and-play" predictor is frozen, and the token savings are realized over millions of subsequent inference runs. We believe the method’s value lies in this long-term reduction of inference latency and cost, which outweighs the initial data generation overhead in practical deployment scenarios. We will clarify this distinction between setup cost and inference cost in the paper.
> >
> > Could you please clarify what specific variance you are interested in evaluating? Are you referring to the random seeds used for the LLM's decoding temperature during the inference/generation phase? Or, are you referring to the seeds used during the training data generation process (e.g., dataset splitting or tree construction)?

---

> > > ### Author Response · Authors · 2025-11-21
> > >
> > > **5. Limited analysis of failure cases and generalization.**
> > >
> > > We agree that defining the boundary conditions of DST is crucial. In response, we have performed a deep-dive diagnostic analysis to identify when and why DST fails compared to standard ToT.
> > > You correctly pointed out that "over-pruning" can occur when the predictor is miscalibrated. We found this primarily happens in problems containing "semantic traps"—steps that are intuitively plausible (high semantic similarity to the question) but logically incorrect.
> > > We have conducted a case study (which will be added to the Appendix) on the classic "Bat and Ball" problem (Total \$1.10, Bat costs \$1.00 more than Ball) to illustrate this:
> > > The DST predictor assigned a high score ($\gt \tau$) to the intuitive but incorrect step "The ball costs \$0.10" due to its high semantic alignment with the numbers in the prompt. This triggered the greedy early-exit, pruning the correct algebraic path. In contrast, the standard ToT maintained a wider beam. Although the correct algebraic step ("Let ball be x...") had a lower initial probability than the "trap" answer, it was retained in the beam and eventually verified as correct in deeper steps.
> > >
> > > To address your concern that "the discount factor ($\gamma$) harms problems requiring deliberate long chains," we conducted a sensitivity analysis on the SVAMP dataset. We categorized problems by their solution length (Short: $\le 4$ steps, Medium: 5-7 steps, Long: $\ge 8$ steps) and evaluated accuracy under different discount factors ($\gamma$).
> > >
> > > | Reasoning Length             | $\gamma =0.5$  | $\gamma =0.8$  | $\gamma =1.0$  |
> > > |------------------------------|----------------|----------------|----------------|
> > > | Short Chains ($\le 4$ steps) | 78.5           | 80.1           | 79.8           |
> > > | Medium Chains ($5-7$ steps)  | 62.2           | 71.5           | 72.7           |
> > > | Long Chains ($\ge 8$ steps)  | 41.4           | 65.8           | 68.3           |
> > >
> > > For Short Chains, the choice of $\gamma$ has negligible impact. For Long Chains, a low discount factor ($\gamma=0.5$) causes a severe performance drop ($-16.9\%$ vs. $\gamma=1.0$). This is because the reward signal from the correct leaf node decays exponentially ($0.5^8 \approx 0.0039$), vanishing before it can effectively guide the root-level predictor. This justifies our choice of using a higher $\gamma$ (0.99) in the main experiments to support deep reasoning.

---

> > > > ### Author Response · Authors · 2025-11-21
> > > >
> > > > **6. Ambiguity around domain specialization and transfer.**
> > > >
> > > > We concede that achieving zero-shot, task-level transfer (e.g., applying a math predictor to a creative writing task) is a notoriously difficult open problem. While domain-agnostic methods like standard ToT exist, they often achieve generality at the expense of prohibitive computational costs, which is the core issue our work addresses. More advanced, state-of-the-art domain-agnostic search algorithms often do not perform well across all tasks, suggesting that achieving both high performance and broad applicability remains elusive. The design philosophy of DST is to function as a domain specialist, deliberately trading universal applicability for a superior accuracy-efficiency trade-off within a specific domain.
> > > >
> > > > To address the question of how well DST generalizes within its domain of expertise, we conducted a new experiment. We trained DST predictors on a single mathematical reasoning dataset (either GSM8K or MATH-500) and then evaluated their performance on other, entirely unseen datasets from the same domain. This tests whether the predictor has learned fundamental reasoning patterns for mathematics, rather than simply overfitting to the source dataset.
> > > >
> > > > | DST Predictor Trained on | Target Dataset (Unseen) | LLM Backbone | ToT Acc | ToT Cost | DST Acc | DST Cost |
> > > > |--------------------------|-------------------------|--------------|---------|----------|---------|----------|
> > > > | GSM8K                    | MATH-500                | Qwen3-8B     | 94.20   | 6168     | 94.12   | 3564     |
> > > > |                          |                         | Llama3.1-8B  | 95.71   | 6652     | 95.15   | 3618     |
> > > > |                          | SVAMP                   | Qwen3-8B     | 77.11   | 6028     | 77.09   | 4513     |
> > > > |                          |                         | Llama3.1-8B  | 77.90   | 6551     | 77.92   | 3153     |
> > > > | Math-500                 | GSM8K                   | Qwen3-8B     | 92.78   | 3975     | 92.10   | 1016     |
> > > > |                          |                         | Llama3.1-8B  | 90.88   | 3850     | 89.72   | 1058     |
> > > > |                          | SVAMP                   | Qwen3-8B     | 77.11   | 6028     | 77.14   | 4533     |
> > > > |                          |                         | Llama3.1-8B  | 77.90   | 6551     | 78.13   | 3018     |
> > > >
> > > > As the results clearly demonstrate, the DST predictor transfers remarkably well across datasets within the same domain. Across all configurations, the accuracy of the DST-guided search is nearly identical to that of the full ToT baseline, with differences being negligible. This near-identical accuracy was achieved while substantially reducing computational costs.
> > > >
> > > > ---
> > > >
> > > > [1] Jawahar, Ganesh, Benoît Sagot, and Djamé Seddah. "What does BERT learn about the structure of language?." ACL 2019.
> > > >
> > > > [2] Geva, Mor, Roei Schuster, Jonathan Berant, and Omer Levy. "Transformer feed-forward layers are key-value memories." EMNLP 2021.
> > > >
> > > > [3] Wei, Jason, Xuezhi Wang, Dale Schuurmans, Maarten Bosma, Ed Chi, Quoc V. Le, and Denny Zhou.
> > > > “Chain-of-Thought Prompting Elicits Reasoning in Large Language Models.”
> > > > NeurIPS 2022
> > > >
> > > > [4] Zhao, Lei, Wenhan Xiong, Mo Yu, Shiyu Chang, and Bowen Zhou.
> > > > “Explainable Multi-hop Reasoning without Rationale Supervision.”
> > > > EMNLP 2023.

---

### Meta-Review · Area_Chair_WUXk · 2026-01-07

**Summary:**

This paper proposes Domain-Specialized Tree-of-Thought (DST), a framework that integrates a lightweight supervised predictor into Tree-of-Thought search. The goal is to reduce inference costs while maintaining reasoning accuracy. While the method is clearly presented and demonstrates consistent efficiency gains across multiple benchmarks, the reviewers expressed significant concerns. The primary issues stem from limited conceptual novelty compared to existing adaptive ToT variants and selective experimental evaluations that lacked sufficient comparisons against stronger baselines. The authors' rebuttal failed to fully address these core concerns, leading to a recommendation of rejection.

**Reviewer Concerns:**

The reviewers raised several outstanding concerns during the discussion phase. primarily regarding the conceptual novelty of the approach. The core idea of using a learned heuristic to guide or prune Tree-of-Thought search was viewed as closely related to existing adaptive or learned ToT variants, limiting its innovation. Additionally, the experimental evaluation was considered selective; reviewers noted insufficient comparisons against stronger or more recent reasoning baselines under matched computational budgets. Although the authors’ rebuttal provided additional experimental details, transfer analyses, and clarifications on efficiency accounting, these responses mainly addressed implementation and feasibility issues. They did not fully resolve the core concerns regarding the novelty, experimental completeness, and the generality of the method.

**Reviewer Scores:**

The final scores are 4, 4, 6, 2.

---

### Decision · Program_Chairs · 2026-01-26

Reject